# Toward Synthetic Physical Fingerprint Targets

**DOI:** 10.3390/s24092847

**Published:** 2024-04-29

**Authors:** Laurenz Ruzicka, Bernhard Strobl, Stephan Bergmann, Gerd Nolden, Tom Michalsky, Christoph Domscheit, Jannis Priesnitz, Florian Blümel, Bernhard Kohn, Clemens Heitzinger

**Affiliations:** 1Austrian Institute of Technology, 1210 Vienna, Austria; bernhard.strobl@ait.ac.at (B.S.); bernhard.kohn@ait.ac.at (B.K.); 2Bundesamt für Sicherheit in der Informationstechnik, 53175 Bonn, Germany; stephan.bergmann@bsi.bund.de (S.B.); gerd.nolden@bsi.bund.de (G.N.); 3IDloop GmbH, 07745 Jena, Germany; t.michalsky@idloop.com (T.M.); c.domscheit@idloop.com (C.D.); 4Hochschule Darmstadt, 64295 Darmstadt, Germany; jannis.priesnitz@h-da.de; 5Biometrie-Evaluations-Zentrum (BEZ) Hochschule Bonn-Rhein-Sieg, 53757 Sankt Augustin, Germany; florian.bluemel@h-brs.de; 6Institute of Information Systems Engineering/Research Unit of Machine Learning, Technische Universität Wien, 1040 Vienna, Austria; clemens.heitzinger@tuwien.ac.at

**Keywords:** biometric, fingerprint, sensor, synthetic, phantom, standardization, validation

## Abstract

Biometric fingerprint identification hinges on the reliability of its sensors; however, calibrating and standardizing these sensors poses significant challenges, particularly in regards to repeatability and data diversity. To tackle these issues, we propose methodologies for fabricating synthetic 3D fingerprint targets, or phantoms, that closely emulate real human fingerprints. These phantoms enable the precise evaluation and validation of fingerprint sensors under controlled and repeatable conditions. Our research employs laser engraving, 3D printing, and CNC machining techniques, utilizing different materials. We assess the phantoms’ fidelity to synthetic fingerprint patterns, intra-class variability, and interoperability across different manufacturing methods. The findings demonstrate that a combination of laser engraving or CNC machining with silicone casting produces finger-like phantoms with high accuracy and consistency for rolled fingerprint recordings. For slap recordings, direct laser engraving of flat silicone targets excels, and in the contactless fingerprint sensor setting, 3D printing and silicone filling provide the most favorable attributes. Our work enables a comprehensive, method-independent comparison of various fabrication methodologies, offering a unique perspective on the strengths and weaknesses of each approach. This facilitates a broader understanding of fingerprint recognition system validation and performance assessment.

## 1. Introduction

Fingerprint recognition has emerged as a pivotal technology with broad applications spanning law enforcement, national security, mobile device authentication, and access control. Its reliability is underscored by the persistence of fingerprints over time, as evidenced by studies such as Monson et al. [1] and Yoon et al. [2]. Coupled with its high accuracy and cost-effectiveness [3], fingerprint recognition stands as a dependable biometric identifier.

However, ensuring the quality and consistency of fingerprint recognition systems presents significant challenges, particularly in calibrating and standardizing fingerprint sensors. Quality encompasses factors crucial for precise recognition such as imaging fidelity, uniform distribution of gray values, contrast transfer function, and more. Yet, evaluating these systems solely based on technical specifications overlooks factors such as fingerprint smearing during rolling or image stitching errors, necessitating a more comprehensive assessment approach incorporating a fixed ground truth for comparison with the recordings.

Two main measurement methodologies are commonly employed for evaluating fingerprint recognition systems. The first method involves human sampling, where real users are recorded using a previously certified device for comparative analysis [4]. While this approach allows for the measurement of the quality features mentioned above, it suffers from selective bias in participant selection, rendering it non-repeatable, and misses a ground truth to compare to.

The second option utilizes standardized targets, which serve as benchmarks for sensor optimization and performance assessment [5]. These targets evaluate various technical aspects of the sensor [6], and certain targets are designed to generate fingerprint-like patterns, commonly referred to as phantoms in the biometric community. These targets can be tailored to represent a diverse range of users while adhering to a known ground truth for comparing recordings. Moreover, finger-like targets can replicate real-world scenarios, such as fingerprint smearing during rolled recordings. This comprehensive approach ensures consistent quality assessment of the sensor’s behavior across different scenarios.

When standardizing a fingerprint sensor, there are currently two internationally recognized standards to follow, along with various national standards such as the German Technical Guidelines [7]. The widely used international standards include the Personal Identity Verification (PIV) procedures, catering to single-finger capture devices utilized for person verification [8], and the Appendix F standard [9], which qualifies devices for large-scale applications in person identification. Both standards employ different standardized targets, including Ronchi, sine wave, and other patterns. Additionally, to assess fingerprint image quality, device vendors must supply a set of recorded fingerprint images.

Similar to the evaluation of fingerprint recognition systems, the process of selecting participants for testing fingerprint sensors may introduce biases and fail to accurately represent the diverse range of individuals encountered in real-world scenarios. One solution to this issue is the use of standardized phantoms that mimic the features of real users, allow for precise comparisons against a ground truth, and at the same time cover real-world use cases while adhering to data privacy constraints. This approach necessitates the large-scale creation of realistic synthetic phantoms, which serves as the motivation for this work.

The objective of this research is to innovate and evaluate various novel phantom manufacturing technologies comprehensively, focusing on their quality, cost-effectiveness, and suitability. This unique approach of comprehensivly exploring and assessing diverse methodologies facilitates the integration of knowledge across manufacturing techniques and allows a straightforward comparison of the different approaches. Consequently, these insights can be leveraged to fabricate high-quality phantoms tailored to the specific recording modality, thereby enhancing the efficacy of fingerprint sensor validation and in the future maybe also standardization.

Furthermore, generating substantial training data for machine learning models stands as another critical motivation for this research. Synthetic targets provide reproducibility and controllability, facilitating the creation of large datasets for effective training of machine learning algorithms. This, in turn, can bolster the development of more robust and accurate fingerprint recognition models. However, achieving this necessitates a scale-able, precise, and cost-effective production methodology for physical phantoms. An important caveat in this endeavor is the challenge of ensuring diversity within the dataset to encompass the full spectrum of possible fingerprints.

An additional imperative driving the need for synthetic targets is the inherent challenge of privacy and data security associated with the use of real fingerprint datasets. Synthetic targets alleviate concerns by providing a means to generate realistic fingerprint data without compromising the privacy of individuals.

The work is structured in the following way: First, we present related work regarding both synthetic fingerprint creation and physical fingerprint generation in Section 1.1. Next, we describe the tools used to generate 2D synthetic fingerprints in Section 2, followed by the detailed explanation of the different physical production methodologies in Section 3, wherein we describe laser engraving in Section 3.1, 3D printing in Section 3.2, and CNC machining in Section 3.3. After the methods sections, we present the results of generating the 2D synthetic fingerprints in Section 6, the results of the physical processes creating the phantoms in Section 7, and finally the results of measuring the performance of those phantoms in Section 8. A discussion section in Section 9 follows, and this work ends with a conclusion in Section 10.

### 1.1. Related Work

#### 1.1.1. Synthetic Fingerprint Generation

In recent years, synthetic fingerprint generation has garnered substantial attention from researchers aiming to overcome challenges associated with small, proprietary databases and privacy concerns in fingerprint recognition research.

The necessity for large datasets in fingerprint recognition is crucial. Large datasets facilitate the training of models for matching fingerprints and enable more robust validation of results. Moreover, the diversity within these datasets is crucial for capturing the variability present in real-world scenarios, thereby ensuring the generalizability of developed algorithms.

Several notable publications have focused on generating synthetic fingerprint images using various algorithms and techniques [10,11,12,13,14,15]. However, a critical distinction lies in the fact that these existing publications primarily concentrate on simulating fingerprint images rather than physically creating synthetic fingerprint targets. To our knowledge, there has been little research conducted on the physical creation of synthetic 3D fingerprint targets, making this paper a pioneering endeavor in elucidating the intricacies of this process.

The motivation for the biometric community to explore synthetic fingerprint generators stems from inherent limitations in current fingerprint recognition systems, particularly the scarcity of publicly available fingerprint datasets. Well-known datasets, such as FVC [16], LivDet [17], and National Institute of Standards and Technology (NIST) Special Dataset 302 [18], present challenges due to their restricted identities and impressions per finger, hindering comprehensive research on critical topics such as training deep networks for fixed-length fingerprint embeddings [10,19]. Furthermore, the uncertain availability of the currently existing public datasets, marked by removals due to privacy regulations, adds another layer of complexity to fingerprint recognition research.

Pioneering the field, Cappelli et al. introduced SFinGe, a synthetic fingerprint generation approach that automates the creation of extensive fingerprint databases, crucial for the effective training, testing, optimization, and comparison of fingerprint recognition algorithms [12]. SFinGe emulates images acquired with electronic fingerprint scanners, providing a valuable resource for evaluating algorithms in online acquisition scenarios.

The SynFi approach by Riazi et al. [11] introduces a novel method utilizing both Generative Adversarial Networks and Super-Resolution techniques for automatic high-fidelity synthetic fingerprint generation at scale. By emphasizing computational indistinguishability from real fingerprints, SynFi advances the realism of synthetic datasets, presenting a significant contribution to the field.

The Clarkson Fingerprint Generator, developed by Bahmani et al., is a Generative Adversarial Network-based tool for generating synthetic fingerprints [20]. Their approach addresses the shortage of publicly available, large-scale fingerprint datasets. Employing a multi-resolution and progressive growth training approach, the Clarkson Fingerprint Generator ensures the generation of high-quality fingerprint images. Moreover, the study demonstrates the model’s robustness by ensuring that no training information is leaked through rigorous matching of every synthetic fingerprint with bonafide ones.

PrintsGAN, proposed by Engelsma et al., has been instrumental in synthesizing a database of 525,000 fingerprints, each associated with multiple impressions [10]. A crowd-sourced evaluation attests to the realism of the dataset, showcasing improved performance on real fingerprint datasets, specifically in extracting fixed-length fingerprint representations.

Makrushin et al. employ a different approach [14], using minutiae patterns and basic fingerprint types with a pix2pix network to generate synthetic fingerprint data. By reconstructing fingerprints from pseudo-random minutiae and leveraging pix2pix networks, this work addresses privacy concerns while compensating for missing sample variability in real biometric datasets.

The SynCoLFinGer method, proposed by Priesnitz et al. [15], extends the landscape of synthetic fingerprint generation by introducing the capability to generate contactless fingerprint images. Unlike existing methodologies that primarily focus on simulating traditional contact-based fingerprints, SynCoLFinGer meticulously models and applies constituent components, including capturing conditions, subject characteristics, and environmental influences, to a synthetically generated ridge pattern using the SFinGe algorithm.

#### 1.1.2. Physical Fingerprint Generation

Building upon the advances in and methodologies for synthetic fingerprint generators, it is essential to address the challenges and intricacies associated with the physical synthesis of fingerprint targets. While synthetic fingerprint generation has made significant strides in simulating intricate patterns and characteristics of human fingerprints and the mechanical understanding of human skin has matured [21,22], transitioning from digital simulations to tangible physical entities introduces numerous challenges.

Achieving synthetic skin that mimics human skin’s elasticity and tactile properties remains crucial for realistic touch-based fingerprint replication. Recent research highlights potential materials, including gelatinous substances and elastomers, offering avenues to address these challenges [23,24,25,26].

Additionally, in recent years, remarkable advances in the field of 3D printing have been made, revolutionizing various industries with its capabilities [27,28]. Notably, the evolution of low-cost 3D printers has democratized access to this technology, making it more accessible to researchers in different disciplines [29].

In tandem with 3D printing, laser engraving has emerged as a powerful industrial technology suitable for target creation [30,31]. In the context of fingerprint Presentation Attack Detection (PAD) using synthetic ridge patterns, laser engraving has already proven to be an effective tool [32]. In addition, more and more cost-efficient systems are available on the market. Similar to laser engraving, CNC machining is another process that is able to create precise target masters [33].

Expanding the arsenal of techniques for physical synthetic fingerprint generation, recent research introduces a novel approach using unconventional polycarbonate molding [34]. This method involves pressing human fingerprints onto solvent-softened plastic plates, such as polycarbonate chips, followed by casting with polydimethylsiloxane, a popular elastomer. The result is a flexible, micrometerscale-detailed reproduction of the fingerprint. These phantoms retain the exact 3D features of the ridge structure of the fingerprint but do not keep the general 3D shape of the finger. This innovative bench-top method offers a practical and instrumentation-free means of mass reproducing microstructures with high fidelity but lacks the overall 3D geometry of the finger and is limited to replicating real human fingerprints.

Another avenue of research focuses on projecting bonafide 2D fingerprint patterns with known characteristics onto a generic 3D finger surface using a state-of-the-art 3D printer [5,35,36]. The resulting physical 3D targets are fabricated with material similar in hardness and elasticity to human finger skin and are worn on a finger or above the hand. Experimental results demonstrate that these fabricated 3D targets can be imaged using commercial optical as well as capacitive fingerprint readers. Additionally, the salient features in the 2D fingerprint patterns are preserved during the synthesis and fabrication of 3D targets. Importantly, the intra-class variability between multiple impressions of the 3D targets captured using the optical fingerprint readers does not degrade the recognition accuracy. However, the introduction of wearable fingerprint targets poses challenges related to repeatability when calibrating and evaluating fingerprint sensors, as it introduces the human element into the process chain. The intricacies of human-induced factors such as contact pressure, angle, and position present difficulties in precise control, hindering the automated upscaling of the calibration and evaluation process.

Moreover, the generation of physical fingerprints is closely linked to the creation of presentation attacks. In this context, artificial replicas known as Presentation Attack Instruments (PAIs) are utilized to deceive fingerprint capturing devices by mimicking fingerprint characteristics. These replicas can be crafted from a variety of materials such as gelatin, silicone, Body Double, different types of glue, or latex [37,38,39,40]. When targeting contact-based fingerprint capturing devices, it is essential that the PAI replicates the elasticity of real skin. Additionally, the PAI must meet criteria for persistence, allowing it to be used multiple times while also being easy to assemble and cost-effective. Silicone and Body Double, in particular, are well suited to fulfill these requirements [40].

In light of these challenges, this paper endeavors to navigate the complexities of physical synthetic fingerprint production, exploring innovative methodologies and approaches to overcome inherent limitations. By addressing these challenges head-on, we aim to contribute significantly to the standardization, reliability, and efficacy of fingerprint recognition systems, fostering advances in both synthetic and physical realms of fingerprint technology.

## 2. Methods—Generating Synthetic Fingerprints

The production of synthetic fingerprints is essential for generating openly available phantoms, and it has been accomplished utilizing the Synthetic Fingerprint Generator (SFinGe) [12]. SFinGe is a software tool specifically designed to synthesize realistic fingerprint images. The synthetic fingerprints produced in this study aimed to encompass the full spectrum of major finger classes, including left loops, right loops, arches, tented arches, and whorls.

The acquisition parameters in the SFinGe tool were set at a standardized 21.1 mm × 28.4 mm dimension with a resolution of 500 dots per inch (DPI). The fingerprint mask utilized maximal values to ensure the largest possible fingerprint representation. Pores were incorporated into the synthetic fingerprints, and the ridge density was configured to its highest setting. Importantly, no scratches were introduced into the synthetic fingerprints during the generation process.

To optimize the character of the synthetic fingerprints, noise and prominence in the ridges were minimized. Additionally, no background was incorporated into the synthetic fingerprints, maintaining focus on the fingerprint patterns themselves.

After the synthetic fingerprint images were created, we needed to increase the image resolution since the manufacturing methods are more precise than 500 dpi or even 1000 dpi. For this, we first enhanced the image using a Gabor filter bank consisting of 8 orientation filters, followed by a threshold-based binarization. In the next step, we converted the image into a vector graphic by using the path tracing capabilities of Inkscape. The resulting image can be used to generate fingerprint phantoms with an arbitrary image precision.

The process is depicted in Figure 1 using a left loop fingerprint as an example. The initial output of the SFinGe program, configured with the settings described previously, is presented in the first figure (Figure 1a). Subsequently, in the following step illustrated in Figure 1b, the results of applying the Gabor filter are displayed. This filter serves to close small gaps that simulate missing parts of a ridge line, arising from variations in ridge line height and an uneven pressure distribution in real fingerprints. Next, Figure 1c showcases the binarized image where gray areas are converted to either ridge lines or ridge valleys. The result of the final path tracing step, where the image is converted to a vector graphic, can be seen in Figure 1d.

## 3. Methods—Generating Synthetic 3D Targets

This section outlines the diverse methods employed to generate synthetic 3D phantoms. From laser engraving, a technology renowned for its precision in replicating fine details, to advanced 3D printing techniques, and finally to CNC machining, each method is meticulously explored. The subsequent subsections delve into the details of the material-specific processes.

### 3.1. Laser Engraving

Laser engraving offers a precise and controlled method for ablating intricate patterns, closely mimicking the unique ridges and details of human fingerprints. This technology excels in reproducing fine details, allowing for the creation of high-fidelity physical fingerprint targets and molds. It was previously utilized by [32] to create a flat mold that can be filled with various materials.

For our process of creating phantoms, two different methodologies were tested. We instructed the company Seprinto (https://seprinto.com/laserline-secure/, accessed on 19 March 2024) to laser engrave the target structures directly into elastomer plates. Second, the company Pulsar Photonics (https://www.pulsar-photonics.de/, accessed on 19 March 2024) was instructed to use an aluminum half-pipe as an intermediary step to create a 3D mold, which we then filled with silicone, but it could be filled with an arbitrary filling substance to create the phantom. This novel method of creating a 3D mold using laser engraving has, to the best of our knowledge, not been done before for fingerprint phantom production.

The direct laser method for imprinting fingerprint patterns onto flat materials costs approximately €200 per fingerprint phantom. When considering the aluminum half-pipe, the cost includes both setup expenses to configure the machinery and manufacturing costs for repeated production. The setup fee amounts to roughly €8000, with a manufacturing cost of about €1000 per fingerprint phantom.

#### 3.1.1. Elastomer Target

To assess the suitability of a specialized elastomer for fingerprint target creation, two test samples with an early version of the synthetic fingerprint were lasered from plates also provided by Seprinto, a company known for producing printing plates for banknotes using this material. The lasering process involved specifying a penetration depth of 100 µm, and the laser operated at a resolution of 5080 dpi. The elastomer plates were delivered in two distinct qualities, the Laserline EPDM CSX 5K with a thickness of 0.95 mm (78 Shore A hardness) and the thicker Laserline EPDM CSC with a thickness of 1.42 mm (76.5 Shore A hardness), both with polyester support material.

These elastomer plates were selected based on their varying thickness and hardness properties to investigate their impact on the quality and robustness of the ridge line pattern. The early version of the synthetic fingerprint was only binarized but not upscaled, constraining the resulting phantom to a resolution of 500 dpi. The result of the laser engraving can be seen in Figure 2a.

After the elastomer plates were lasered to create the fingerprint target master, the next step involved preparing them for use on fingerprint sensors. To facilitate this, the elastomer targets were cut from the stripe, carefully bent, and then applied to a cylindrical wooden target holder. Because of the elasticity of the elastomer material, the bend is only temporary, and when removing the targets from the target holder, they transition back into their flat shape. The wooden holder, with a diameter of 20 mm, served as a stable platform for the elastomer targets, ensuring consistent and repeatable placement during the evaluation process. The result can be seen in Figure 2b.

#### 3.1.2. Silicone Target

In addition to the elastomer targets, two different silicone plates were used as a substrate for the fingerprint target creation by directly lasering the target fingerprint into the silicone plates. The first was purchased from the company Gospire, and the laser-engraved phantom is shown in Figure 3a. It is a 1.8 mm thick silicone training skin for tattoo artists, and the second, which is shown in Figure 3b, was manufactured in-house.

For the in-house creation of the silicone plate, we used the Dragon Skin 10 Fast silicone and Silc Pic color pigments to achieve a light skin tone, as outlined in the manufacturer’s process. After the silicone was mixed and degassed using a vacuum chamber, it was poured onto a glass plate, with additional spacers positioned to determine the thickness (also 1.8 mm). A second glass plate was used to press down on the silicone before the curing process to create a flat surface with consistent depth. In order to ensure a uniform and constant distance between the two glass plates during the curing process, the second glass plate was weighted. The Dragon Skin 10 Fast silicone used in this process has a processing time of approximately 20–25 min and cures after about 4 h.

Due to the airtight seal formed between the glass plates during compression of the silicone, separating the plates can be challenging. To facilitate the separation, we used compressed air. The procedure involves directing a compressed air gun to a point where silicone seeps out from the gap between the plates, attempting to introduce air between the plates and the silicone. This technique simplifies the detachment process, ensuring the integrity of the silicone plates for subsequent applications.

#### 3.1.3. Aluminum Half-Pipe Mold

An alternative method for generating high-precision fingerprint samples is to create a negative mold using an aluminum tube. The inside of the tube bears the negative image of a fingerprint, engraved using a laser, allowing the reproduction of detailed fingerprint patterns. The aluminum tube, specifically an EN-AW 6063 alloy, was partly cut into halves with a length of 300 mm, an outer diameter of 23 mm, and an inner diameter of 20 mm. Approximately 20 mm of the complete tube remained at one end to facilitate gripping by the rotating device of the laser during engraving, as illustrated in Figure 4a. Note that the tube had to be sanded down to a little less than 180° to allow for the laser to reach the edge of both sides.

To create the mold, a plug can be inserted into the tube to create the rounded tip and cylindrical finger shape. This plug can be 3D printed using cheap, conventional 3D resin printers. During casting of the silicone, the plug is inserted on one side and secured on the other side with a rubber band, with the filling openings for the casting material located on the top. See Figure 4b for a rendering of the plug.

### 3.2. 3D Printing of the Resin Mold

A different processing approach involving the 3D printing of a mold was performed, as done by [36]. But, instead of printing the 3D scan of a human finger, as done by Engelsma et al., our novel approach was to design and print a cylindrical finger shape with a consistent rolling behavior and the ability to project an arbitrary, synthetic fingerprint pattern onto it. This method centers around creating a 3D model of the mold, with a key step being the delineation of ridge lines in the fingerprint pattern. This is accomplished by projecting a ridge line texture onto a cylindrical 3D structure, featuring a spherical apex. Then, vertices on the elevated segments of the ridge line hills are extruded along their face normals, resulting in the formation of the 3D ridge line pattern. To achieve this, we employ the ABF++ algorithm [41], integrated into the 3D modeling software Blender version 4.0 (https://www.blender.org/, accessed on 19 March 2024) to project the 2D fingerprint texture onto the 3D finger mesh. It is worth mentioning that this was the sole method where we could project the fingerprint pattern onto the rounded fingertip.

When creating a mold, it is critical to ensure a seamless separation process between the mold and the final phantom, necessitating the refinement of ridge lines and avoidance of pixel rasterization to mitigate adhesion at sharp edges. This optimization entails upscaling the fingerprint texture (see Section 2 or Section 6) and the introduction of a secondary texture into the extrusion process. This second texture is an identical ridge line texture with added Gaussian blur (kernel size of 12×12 pixels) to create a smooth ridge line edge.

The mold architecture is designed as a two-part system, with each component housing half of the finger bisected along the longitudinal axis. This design facilitates the amalgamation of disparate components, yielding a fingertip phantom endowed with a different fingerprint pattern on both the anterior and posterior aspects of the digit. This allows us to create two distinct fingerprint recordings using a single phantom that is rotated.

Additionally, the mold encompasses two apertures on the side distal to the fingertip, serving as ingress points for silicone infusion. Positioned between these apertures is the target holder fixture, affording secure immobilization to the target holder. This target holder can be grabbed by a robotic arm or human operator. This fixture preserves the central positioning of the target holder within the final fingerprint target, thereby mitigating variations in finger-to-bone distance and ensuring uniform elasticity.

Note that the processing of a mold instead of a target directly allows for a variety of different filling materials. We considered both Gelafix Kryolan as well as silicone.

Our study additionally encompasses the evaluation of two distinct 3D printing methods. Firstly, we harnessed the in-house ES2 Elegoo Saturn 2-8K (produced by Elegoo, Shenzhen, China) resin printer and leveraged Masked Stereolithography (MSLA). In parallel, the Stereolithography (SLA) printer service offered by Alpine3D GmbH constituted the second modality. While both MSLA and SLA constitute stereolithography approaches utilizing a vat of photo-reactive liquid resin selectively exposed to light, their difference lies in the light source employed. SLA adopts a precision laser for resin curing, whereas MSLA relies on a digital screen, typically an LCD, for the selective solidification of liquid resin into the desired 3D configuration.

The cost of the 3D resin printing of the fingerprint mold differs between the two options. Using the in-house ES2 Elegoo Saturn 2-8K resin printer leads to a material cost of under one euro per mold and an acquisition cost of around 500€ for the printer. Using the printing service of Alpine3D costs 3000€ for configuration of the sensor and 50€ per mold afterward.

The results demonstrate the performance of the ES2 Elegoo Saturn 2-8K resin printer with an early rendition of the finger mold model (Figure 5a) and the Alpine3D GmbH (Tirol, Austria) SLA service employing the finalized mold model (Figure 5b). Refinements in the latter iteration include a heightened uniformity in ridge length depth at the border regions, an adaptable mold half-connection framework facilitating arbitrary concatenation, and an augmented depth in the target holder fixture to preclude potential artifacts arising from liquid entrapment in the fingertip mold. Noteworthy is the transparent manifestation of the final mold, mitigating undesired interactions between color particles in the resin and the silicone filling.

During mold creation, limitations in the acquisition area of the synthetic fingerprint at a resolution of 500 dpi restrict the effective area of the phantom with a fingerprint pattern. To enable the testing of a full nail-to-nail unrolling scenario, the full fingerprint area is utilized by mapping the synthetic fingerprint image to the fingerprint master, disregarding scaling. The spherical tip of the target master necessitates stretching in the mapping process, as conformality and equiareality cannot be simultaneously maintained, as described by Carl Friedrich Gauß in his Theorema Egregium [42] (pp. 507–509). The ABF++ algorithm of [41], based on angle-based flattening, is employed for 2D-to-3D mapping, resulting in minimal stretching.

To address the sensitivity of template comparison algorithms to scaling changes, fingerprint target measurements were conducted and recordings scaled accordingly. This approach not only eliminates variability in template comparison scores due to different shrinkage behaviors of various filling materials but also ensures accurate and consistent evaluations for the projected fingertip area. See Section 5.2.1 for more details.

#### 3.2.1. Filling—Kryolan Gelafix

Gelafix is a special effects material used to create skin effects and prosthetics produced by Kryolan (https://de.kryolan.com/, accessed on 19 March 2024). It is a skin-friendly material designed to simulate severe burns and other lesions. Gelafix Skin, for example, is breathable and offers a natural appearance for actors. In the context of crafting fingerprint phantoms, Gelafix serves as a dynamic filling material within the mold structure. It has to be heated until it melts, in order to use it as a liquid filling material. In practice, this can be done using a microwave for a few seconds or other heat sources like an oven or a water bath.

#### 3.2.2. Filling—Silicone

We utilized Dragon Skin silicons, a platinum-cure liquid composition known for its application in various fields, including movie special effects, production molds, medical prosthetics, and anthropomorphic robotic hands [43].

Table 1 outlines key properties of Dragon Skin 10 Fast, Dragon Skin 20, (produced by Smooth-On Inc., Macungie, PA, USA) and human skin, offering a comparative basis for evaluating their suitability [36,44,45,46,47,48].

The Shore A hardness of Dragon Skin 10 Fast is lower than the range of 20–41 observed in human skin [46,47], while Dragon Skin 20 falls at the lower end of this range. The Shore A hardness of the human skin demonstrates a broader spectrum, encompassing the characteristics of the dermis, epidermis, and subcutaneous tissues, contributing to the dynamic nature of the human fingertip. But considering the fingertip itself, the soft fat in the subcutaneous tissue below the hypodermis, results in a softer, more gel-like response than even the softer Dragon Skin variant [43,49].

The preparation of the silicone involves a three-minute mixing process of both silicone components. The Silc Pig silicone color pigments are introduced to enhance the visual authenticity of the silicone phantoms [44]. Following mixing, the mass undergoes degassing in a vacuum chamber for approximately 1–2 min at a pressure of 0.1 bar. This degassing step ensures the removal of air from the mixture, preventing the formation of gas bubbles during the filling process and, consequently, minimizing imperfections in the final fingerprint target.

To address potential biases associated with darker skin tones, the option to incorporate darker skin color pigments is given through the selection of the specific Silc Pig color pigments. This option is more relevant for contactless fingerprint sensors because the influence of skin tones on a fingerprint image captured using fingerprint scanners based on frustrated total internal reflection (FTIR) methods plays a subordinate role.

The release agent, Ease Release 200, is applied from a distance of 20 cm onto the mold and allowed to dry for approximately 30 min. Dimethyl ether/1,1-difluoroethane serves as the solvent carrier for Ease Release 200 [50]. Subsequently, both mold halves are securely fastened with clamps and robust rubber rings. The degassed silicone mixture is then poured into the mold, incorporating a target holder, and left to cure for a duration of 24 h.

We used the Dragon Skin 10 Fast for the filling of the Alpine3D GmbH printed mold and the Dragon Skin 20 for the filling of the previous generation of the mold, printed via the ES2 Elegoo Saturn 2-8k.

A final phantom produced with a silicone filling can be seen in Figure 6.

### 3.3. CNC Machining of the Aluminum Master Target and Filling

This section describes the CNC machining process for master targets, which, to the best of our knowledge, has never been done before for fingerprint phantom production. The process encompasses four distinct phases: milling the aluminum master, surface enhancement through sandblasting, creation of a negative with silicone, and finally filling the negative with soft silicone.

The Fraunhofer Institute for Applied Optics and Precision Engineering (https://www.iof.fraunhofer.de/, accessed on 14 March 2024) was tasked with producing 5 master bodies. All 5 master bodies share an identical basic geometry, manufactured from 20 mm diameter aluminum alloy 6061 round bars using a 5-axis CNC milling machine (HSC 20 linear, produced by Sauer GmbH, Bautzen, Germany) and hyperMILL CAD/CAM software (from OPEN MIND Technologies AG). The patterns on the master targets were a checkerboard structure, vertical Ronchi lines, horizontal Ronchi lines, rings, and the same synthetic fingerprint target as in the first version of the 3D printing process. All targets except for the fingerprint had a period of 1 mm.

The costs of generating further master targets using the setup established creating the 5 master targets is around 1000€ per new master target.

#### 3.3.1. Milling the Aluminum Master

The first stage involved securing the round bar vertically with the back facing upwards. An 8 mm end mill flattened the back surface, followed by milling the outer cylinder to a diameter of 19.2 mm, maintaining a 0.1 mm deviation. The same end mill was used for both operations. Subsequently, a 5 mm end mill was used to create a longitudinal groove within the open pocket on the back. The inner pocket itself was milled using an 8 mm end mill inserted from above. Finally, the outer cylinder was finished to its final diameter of 19.1 mm. The edges of both the open pocket and the back were chamfered to complete this stage. The backside of a target can be seen in Figure 7d. The groove with the open pocket on the back can be used by a robotic arm to grab the targets.

The second stage focused on milling the microstructure. To maintain dimensional accuracy and ensure consistent depth around the cylinder circumference, this stage was performed on the same milling machine and with the same clamping setup as the initial stage. This eliminated the need for re-clamping, which could introduce potential inaccuracies. The size of the micro-tool used was carefully chosen to match the smallest existing structure on the master body: 0.4 mm for Ronchi lines and ringes, 0.2 mm for the fingerprint, and 0.1 mm for the checkerboard.

Before milling the microstructure, we created a horizontal groove (Ronchi line) on a test part with the intended micro-tool. This groove served as a reference for calibrating the feed rate. The depth deviation of the groove from the target was meticulously measured using a microscope (or, alternatively, a stylus instrument). This measured deviation was then incorporated as a feed correction within the path planning software, ensuring precise control during the actual microstructure milling process.

The final stage involved sawing off the round bar on the CNC milling machine. The final hemisphere was then produced on a separate CNC turning machine, utilizing the sawn-off end as the starting point. Furthermore, the other side was also sawn-off to reduce the length of the target master.

The resulting target masters can be seen in Figure 7, where Figure 7a shows the target with the checkerboard pattern, Figure 7b the target with rings, and Figure 7c the target with the synthetic fingerprint.

#### 3.3.2. Surface Enhancement through Sandblasting

Following the milling process, the aluminum master can optionally undergo sandblasting. This technique is employed to refine the surface texture, optimizing it for pattern projection-based 3D sensors and therefore making it directly usable for presenting to contactless fingerprint scanners. The goal is to reach a root mean square deviation of the assessed profile (Rq-ISO 4287-1996) of 1–10 µm. Figure 7e shows the sandblasted Ronchi ring sample.

#### 3.3.3. Creation of a Negative with Silicone

The next step in the replication process is creating a negative mold. We again used Dragon Skin 10 Fast, applying the same preparation technique as described in Section 3.2.2 to ensure a bubble-free cast, except for leaving out the color pigments. First, we used a plastic cylinder with a diameter of 32 mm and a thickness of ≈1 mm that was placed on a solid baseplate. A hot-glue gun was used to create a seal between the base and the cylinder and between the target master and the base. The opening of the master target, which can be used by a robot to grab the target, was sealed off using duct tape. Before pouring the silicone onto the master target, the release agent described in Section 3.2.2 was used to improve the separability between the cylinder and the mold. After filling the mold, the silicone was left to cure overnight.

#### 3.3.4. Creation of a Positive with Silicone

The last phase in the phantom creation process involves filling the negative mold with a soft silicone material. This step results in the creation of the final fingerprint target. We used the same Dragon Skin 10 Fast material that was used for the negative to create the fingerprint phantom, handling it with the same methodology as described in Section 3.2.2. Also, a release agent was used to help with the separability between the two silicons.

## 4. Methods—Measuring Devices

### 4.1. Profilometer

For quality checks of the created fingerprint targets, a Keyence 3D-Profilometer (Control device VR-5000, measure head VR-5200, produced by KEYENCE CORPORATION, 63263 Neu-Isenburg, Germany) was used to capture 3D profile data via structured light illumination technology [51,52]. In addition, the device had different optical and digital magnifications available. For 12× magnification (23.513 µm/pixel), the manufacturer declares a lateral accuracy of 5.0 µm and 2.5 µm for the axial accuracy if the sample is not moved in the *z*-direction to capture the complete target; otherwise, the axial resolution is declared as 4.0 µm. If a 40× magnification (29.583 µm/pixel) is used with the profilometer, the lateral accuracy increases to 2 µm.

Data analysis is conducted using the associated software version 3.3.14.84. Depending on the target under analysis, various processing steps can be executed. For instance, when examining flat silicone samples, the software can compensate for material waviness or unwarp cylindrical targets. However, the profilometers’ software has limitations regarding the unwrapping of cylindrical structures. Instead of fully unrolling the cylinder, the software only corrects the distance from the surface of the underlying cylinder to the sensor. This limitation becomes evident in Section 7.2.2, where the line/structure width for the cylinder in the profile diminishes towards the edges. Had the software accounted for the 3D geometry, the line/structure width should have remained consistent throughout.

Based on the processed images, valuable information regarding structure width, structure depth, edge structure, and surface roughness can be extracted.

#### 4.1.1. Image Processing Workflow

As mentioned before, different samples may require different image processing steps. In the following, the general processing steps are described, followed by a process-specific description:compensation of the general sample tiltcompensation of the sample waviness, e.g., originating from the gluing step, especially for the flat samples or take the geometry of the sample into account, e.g., unwrap/unroll the cylindrical fingerprint targetdefinition of the reference height for relative height measurements

To estimate the material shrinkage while aging, the cylindrical unwarp function of the software was used. By selecting the fingerprint target area, a cylindrical fit over the object was performed, where the user obtains the cylindrical diameter as a parameter.

##### Elastomer Targets

The elastomer targets introduced and described in Section 3.1.1 were detached from the cylindrical target holder for subsequent flattening during profilometer measurements. Initially, a cropping process was employed to eliminate sections of the measurement exhibiting the greatest height disparity, typically observed at the interface between the elastomer and the profilometer stage. Following this, an integrated function was utilized for surface correction, specifically employing the remove the wave option with a correction strength set at 10 out of 20. This procedure ensures the attainment of a flat representation of the elastomer strip while retaining the checkerboard and fingerprint structures intact.

##### Silicone Plate Targets

The silicone plate targets discussed in Section 3.1.2 were affixed to a flat metal sheet to ensure the attainment of as flat a silicone surface as possible. However, in the color-coded height information, indentations arising from the adhesive used for attachment were noticeable. Although these dents were identified, their impact was considered negligible during height profile analysis, which was conducted in a region adjacent to the dents.

Initially, any remaining tilt in the targets was corrected. Subsequently, an integrated function for surface shaping correction, utilizing the remove the wave option with a correction strength set to 20 out of 20, was applied to the 3D profile. This process aims to achieve a flat representation of the silicone plate while preserving the fingerprint structure. Finally, the relative reference height was established to complete the procedure.

##### Finger-Like Targets

To acquire the height of finger-like targets, they were placed unfixed on the profilometer. Since the targets contain a rod-like target holder, an initial tilt adjustment was necessary. In the first step, the targets were aligned along the vertical axis, followed by the application of the implemented cylindrical surface shape correction. Similar to other targets, the relative height of the target was determined visually based on the lowest structure observed on the target.

### 4.2. Greenbit Dactyscan 84c

The Greenbit DactyScan 84c, developed by Thales Cogent (https://www.thalesgroup.com/en, accessed on 19 March 2024), is an optical live fingerprint reader equipped with a large sensor area. Its capabilities extend to the capture of 4-slap and 2-thumb fingerprints, allowing for the collection of all 10 fingerprints using the 4 + 4 + 2 method. The device further facilitates the capture of single rolled and flat fingerprints. For this study, rolled recordings of the phantoms are considered.

The DactyScan 84c has been certified by the FBI as being compliant with the FBI Appendix F standard [53].

It is also compliant with the ANSI standards [54,55] and ISO standards [56].

### 4.3. Microscope

To investigate the surface quality of the material as well as the ridge line density and the ridge line shape, we used the *Vision Engineering Dynascope Stereo Lynx* (produced by Vision Engineering, USA), with which we could reach a pixel resolution of around 3.5 µm/pixel. We used a precise Ronchi calibration target with a 1 mm per cycle line density from the company *Applied Image Inc.* (New York, NY, USA) to calibrate our measurements.

## 5. Methods—Measuring Algorithms

### 5.1. NFIQ 2 Analysis of Phantoms and Synthetic Fingerprints

To measure the visual quality of the synthetic 2D fingerprint images as well as the recorded phantom impressions, we utilized the NIST Finger Image Quality version 2.2.0 (NFIQ 2) as a benchmark [57]. NFIQ 2, a standardized tool, quantitatively evaluates image quality using quality features based on the orientation certainty level, minutiae quality, ridge line uniformity and more, as defined in [58].

It was designed and trained for plain fingerprint recordings. Nonetheless, since plain fingerprint recordings and rolled fingerprint recordings share a similar set of features, we assumed that the NFIQ 2 was indicative of the visual quality also of rolled fingerprint recordings.

We generated NFIQ 2 scores for synthetic images and compared them with rolled impressions from corresponding finger phantoms.

### 5.2. End-to-End Fidelity

Fidelity reflects the degree of the samples’ similarity to its source [59]. In our setting, the generated 2D fingerprint represents the source, and the recorded fingerprint from the corresponding phantom represents the sample. Since the generated synthetic fingerprint images share the same characteristics as real fingerprints, a high end-to-end fidelity indicates that the manufacturing methodologies can be used to create realistic, human-like phantoms.

Furthermore, we decided to test the phantoms shortly after production to avoid distortions in the end-to-end fidelity from varying degradation speeds of the involved materials. Additionally, we investigated the speed and magnitude of material degradation by measuring the shrinkage and hardening of the materials. See Section 7.2.1 and Section 7.2.2 for more.

To comprehensively assess the end-to-end fidelity of the synthetic finger phantoms, we executed a pre-processing procedure to find the optimal scaling values to correct for potential shrinkage and distortions in the 2D to 3D warping process. In the next step, template comparison scores between the 2D synthetic input image and the rolled recording of the fingerprint phantom were calculated. These template comparison scores were calculated using 3 different matchers. More details are given in Section 5.2.2.

#### 5.2.1. Scale Pre-Processing

An important step in preparing the recordings for the matching procedure is to homogenize their scaling, since minor deviations in the scaling can lead to significant changes in their matching performance.

To homogenize scaling, we scaled fingerprint recordings in the x-direction and independently in the y-direction and calculated template comparison scores with the corresponding fingerprint ground truth for each scaling value. In this way, we could quantify the quality of the selected scaling values using the template comparison scores.

We selected the current scaling values using a simple grid search algorithm. For this, we sampled 50 points for scaling along the x-direction and 50 points for scaling along the y-direction, resulting in 2500 possible combinations per image.

The starting points of the grid and the end points of the grid were chosen such that the template comparison score maxima sat roughly in the middle of the grid.

We repeated this grid search for all phantom recordings for all phantom types. In the next step, we selected the 10 scaling values with the highest template comparison scores per recording. These top 10 values were then employed to calculate a weighted average of the corresponding 10 best scaling values for a given recording, where the template comparison scores served as weights. This results in optimal scaling parameters along the x-axis and the y-axis for a given recording. Also, the quality of the identified scaling value could be measured by the average template comparison score of the 10 values used in the calculation. Then, the weighted average of the scaling values over all recordings of a given fingerprint phantom type was calculated, and the quality of the scaling values (mean of template comparison score of the previously taken 10 best scaling values) was used as weights to calculate the average. The resultant x- and y-scaling values were deemed optimal for a given phantom type and were applied to correct the recorded rolled fingerprint samples.

Additionally, we calculated the standard deviation over the quality of the scaling values and averaged over different recordings of the same fingerprint phantom type. A low standard deviation indicated that the template comparison scores for the 10 best scaling values were very similar; therefore, the scaling had less influence on the outcome.

To calculate the template comparison scores, we used the state-of-the-art commercially available automated fingerprint identification system IDkit from Innovatrics [60].

#### 5.2.2. Matcher Comparison

After the optimal scaling values were calculated, we utilized three distinct methodologies to calculate end-to-end fidelity template comparison scores. Firstly, the state-of-the-art commercially available automated fingerprint identification system IDkit from Innovatrics was used [60].

Secondly, the well-established open-source NIST Biometric Image Software (NBIS) distribution, with Mindtct for Minutiae extraction and Bozorth3 for matching, was used [61]. All experiments were conducted using the default parameters without any optimizations, like minutiae quality threshold adaptations.

Finally, a combined open-source fingerprint recognition system was adopted, utilizing FingerNet [62] for minutiae extraction and SourceAFIS [63] for pairing and scoring. It should be noted that the original algorithm uses minutiae quadruplets, i.e., additionally considers the minutiae type (ridge ending or bifurcation). Since minutiae triplets are extracted by the used minutiae extractors, the algorithm has been modified to ignore the type information since the SourceAFIS system does not support this information.

For all three cases, higher score values indicate a high similarity between the rolled recording of the fingerprint phantom and the 2D synthetic input image.

### 5.3. Intra-Class Variability of 3D Target Impressions

Beyond evaluating the end-to-end fidelity, we assessed the intra-class variability of individual fingerprint phantoms. This involved comparing multiple recordings of the same phantom type using the three template comparison tools mentioned in Section 5.2.

Furthermore, we extended the assessment of intra-class variability to different manufacturing methodologies, investigating the phantom interoperability. This includes calculating template comparison scores for rolled recordings of phantoms created using laser engraving on an aluminum half-pipe and comparing them with samples created via laser engraving on silicone samples. Additionally, we determined the intra-class variability of phantoms produced using a 3D-printed resin mold, where we compared silicone against Gelafix as a filling material.

## 6. Results—Generating Synthetic Fingerprints

We created multiple versions of synthetic fingerprints, as can be seen in Figure 8. The first, early version had a restricted image region focused only on the central area around the fingerprint core and can be seen in Figure 8a. It was used in the laser engraving of an elastomer sample. We noticed that for other processing methodologies, we required a larger fingertip section of the fingerprint. Therefore, in the next iteration, we created a fingerprint sample with a larger mask that contained more information on the fingertip, depicted in Figure 8b. This version was used for the laser engraving of both silicone samples and for the laser engraving of the aluminum half-pipe mold. In the next step, we noticed that the core was placed on the spherical part of the fingertip for the case of the 3D-printed mold, which is untypical for a real fingerprint. For the next iteration, we kept the larger mask but chose a lower core position, resulting in Figure 8c. This image was used for the first generation of the 3D-printed resin mold. In the last batch of synthetic fingerprints, we generated one fingerprint for each fingerprint class, except the twin loop type [64], which is often labeled as whorl [65,66]. Those images were used in the second version of the 3D-printed resin mold, and the right loop fingerprint class can be seen as an example in Figure 9a.

The result of the upsampling process can be seen exemplary in Figure 9, which shows the generated fingerprint sample used in the production of the second version of the 3D-printed resin mold with the right loop fingerprint pattern. The resulting vector graphic can be used for arbitrary precision in the production process.

## 7. Results—Generating 3D Targets

### 7.1. Laser Engraving

#### 7.1.1. Elastomer Target

The direct laser engraving of the elastomer material created a flat fingerprint phantom that can be wrapped around a cylindrical finger model to simulate the 3D finger shape. The thinner version of the elastomer material (0.95 mm) behaves preferable when compared to the thicker version (1.42 mm) because of the stiffness of the material, which resists bending around the wooden finger model. Nonetheless, even the thin material was too stiff to create an impression rolling it on the sensor because it required a lot of force to create a visible impression on the sensor. Therefore, we placed the fingerprint phantom flat on the acquisition area on the sensor and used the wooden finger model to put pressure on one part of the phantom. We then rolled the wooden finger model above the phantom, simulating the rolling process.

When analyzing the fingerprint phantom surface using the Vision Engineering Dynascope Stereo Lynx, we noticed that the ridge line thickness appeared to vary locally, as seen in Figure 10c, but investigating this effect with the profilometer, the effect appeared to be an optical one with little impact on the 3D ridge structure.

Examining the laser-engraved early version of the synthetic fingerprint using a profilometer revealed two distinct observations. Firstly, it exposed the limitations of the ground truth data, which were originally utilized at a resolution of 500 dpi, evident in the pixelated edges of the laser-engraved fingerprint. Secondly, it highlighted unnaturally strong width variations within the ridges and valleys. The height disparity between neighboring ridges and valleys measured at 115.6 µm ± 1.2 µm (refer to Figure 11a), while the estimated width of a valley was approximately 304.0 µm ± 19.1 µm (refer to Figure 11b).

Alongside the synthetic fingerprints, a checkerboard pattern is engraved into the elastomer material. By averaging fourteen minimum and maximum measurements, the average depth of the checkerboard is determined to be 121.6 µm ± 1.5 µm (refer to Figure 12a).

The width analysis of the structure revealed variations in the widths of the upper and lower segments as well as the increasing and decreasing shoulders (refer to Figure 12b). Direct comparison between the shoulders indicated that the increasing shoulder measures longer at 160 µm ± 10 µm compared to the decreasing shoulder at 109.4 µm ± 8.5 µm. This discrepancy directly affects the widths of the upper and lower segments of the checkerboard structure. Specifically, the upper plateau appears wider, measuring at 924.8 µm ± 7.4 µm, in contrast to the lower plateau, measuring at 766.8 µm ± 18.2 µm.

Both measurements, conducted on both the checkerboard structure and the synthetic fingerprint, exhibited deviations from the anticipated engraving depth of 100 µm. These deviations surpassed both the error margin of the depth measurement and the axial accuracy of the profilometer under the employed 12× magnification. Another probable source contributing to the disparity from the expected structure depth could be attributed to image processing aimed at rectifying the tilt and waviness within the elastomer stripes.

#### 7.1.2. Silicone Target

The same manufacturing methodology of directly laser engraving the fingerprint pattern into the target material, creating the fingerprint phantom in one step, was tested on silicone plates as well. We used both the Gospire silicone, which is artificial skin created for tattoo artists, and the Dragon skin 10 Fast-based in-house-made silicone plates. Both are approximately 1.8 mm thick. The result of the laser engraving process can be seen in Figure 10d for the Gospire silicone and in Figure 10e for the in-house silicone. The most obvious difference between the two is the coloring of the ridges in the Gospire silicone. This is an effect of the laser, which appears to have burned the surface. Both silicone plates show a very homogeneous ridge thickness and a clear separation of ridge line and valley.

To analyze the silicone plate-based phantoms, measurements with the 3D-Profilometer were performed with the 40× lens and the manufacturer predefined setting for high-resolution acquisitions.

On the Gospire silicone plate over seven nearly parallel laser-engraved valleys, a ten-line-thick profile plot was measured. For each valley, the depth of the valley was measured via the maximum height difference in a hand-selected interval, from the beginning of a ridge to the beginning of the next ridge (see Figure 13a). In total, the mean depth was 86.8 µm ± 2.7 µm. In a similar manner, the valley width for this target was calculated to be 339.7 µm ± 12.8 µm (see Figure 13b).

The in-house manufactured Dragon Skin 10 Fast silicone plate exhibited slight deviations in characteristics, particularly concerning the valley depth and width. The mean depth of valleys for the Dragon Skin 10 Fast silicone plate measured at 63.8 µm ± 2.0 µm (refer to Figure 14a), while the mean valley width averaged 303.1 ± 8.3 µm (refer to Figure 14b). Notably, distinguishing the beginnings and ends of ridges and valleys for the Dragon Skin 10 Fast material proved challenging. Unlike the Gospire silicone plate, the edges in Dragon Skin 10 Fast appeared more rounded, complicating the estimation of structural boundaries (compare profiles in Figure 13b and Figure 14b). Furthermore, a cross-shaped artifact was prominently visible in the Dragon Skin 10 Fast silicone, evident in the color-coded height image of the target introduced during the laser engraving process.

The evaluation of both the in-house created Dragon Skin 10 Fast silicone-based samples and the commercially available Gospire silicone-based sample indicated that the laser engraving method is highly precise overall. For both materials, it was possible to create fingerprint-like structures that closely matched the physiological parameters of fingerprint structures in terms of size.

However, there was a discrepancy in the valley depth of both materials compared to the target depth of 100 µm. This difference may arise from variations in laser power calibration, resulting in different depths in the silicone material aimed at achieving the 100 µm depth. Notably, the two investigated silicones exhibited distinct characteristics when processed by laser engraving, as evidenced by differences in the shapes of the valleys and the more plateau-like ridges observed in the Gospire silicone.

#### 7.1.3. Aluminum Half-Pipe Mold

The second method for crafting fingerprint phantoms via laser engraving employs a two-stage process. Initially, the negative is engraved into a mold material, typically aluminum in our case, which serves as the foundation for producing the final phantoms by filling the mold. Silicone was employed as the filling material in our study. Figure 15a showcases the laser-engraved aluminum mold. The ridge line thickness of the mold precisely mirrors the synthetic fingerprint image, presenting a uniform surface. However, during the engraving of the cylinder, a processing artifact emerged in the form of an offset along a line in the longitudinal finger axis. This offset manifested as a sharp edge in the ridge lines, stemming from variability in the cylindrical aluminum half-cylinder. These artifacts persist in the final fingerprint phantom and are discernible in the captured rolled fingerprints. Figure 15b displays the final fingerprint phantom, illustrating the offset of the ridge lines along a horizontal line in the middle of the image.

For evaluation purposes, the horizontal lines introduced during the laser engraving process are pivotal. Their presence interrupts the ridge lines, generating new minutiae, which can pose challenges for quality and matching processes reliant on minutiae. Nevertheless, it remains feasible to create fingerprint-like structures with a ridge-valley height distance measuring 89.9 µm ± 7.0 µm and a valley width of 364.0 µm ± 20.0 µm.

### 7.2. 3D Printing of Resin Mold

During the creation of the 3D-printed resin mold, we observed the emergence of artifacts. Figure 16a shows an exemplary artifact in the form of a hole. This propagated to the filling process and led to artifacts on the final phantoms. Additionally, further imperfections related to scale and surface shift or tilt may occur. As described in Section 3.2, we adapted the 3D model and outsourced the printing process, resulting in a more precise, reliable, and artifact-free mold form, which can be seen in Figure 16d.

#### 7.2.1. Gelafix Filling

The first material we used to fill the mold with was Kryolan Gelafix, which is available in different colors. It resulted in a fingerprint phantom with a pronounced ridge line structure, which can be seen in Figure 16c. However, an extensive number of small holes covers the fingerprint area, originating from in the material-embedded air bubbles. Those are small enough to not infer with Minutiae based template comparison but could be problematic when considering level 3 features like sweat pores.

After repeated usage of the fingerprint phantom on the fingerprint scanner, we observed a deformation of the fingerprint phantom. Furthermore, extensive shrinkage occurred, resulting in a 17.2% shrinkage in cylinder diameter after 6 months. The measured values can be found in Table 2. This can be seen in Figure 17, where recordings at day 0, day 1, day 4 and after approximately 6 months are shown. Furthermore, Figure 18 shows the profilometer recordings of a fresh Gelafix sample, which can be compared against Figure 19, which shows the profilometer recording of an approximately 6 months old Gelafix sample. In addition to the shrinkage, the elasticity of the material decreased. This meant that rolling the fingerprint phantom on the fingerprint sensor was hampered because only a small area of the fingerprint touched the sensor at the same time.

In addition to the expected overall shrinkage of the cylinder diameter over time, it comes as no surprise that the discrepancy in height between ridges and valleys, as well as the width of the valleys, also diminishes. On the initial production day (Day 0), the average ridge-valley height disparity measured 146.0 µm ± 7.6 µm, with the mean valley width at 329.5 µm ± 17.6 µm. After approximately six months of aging under ambient conditions, the ridge-valley height decreased to 118.1 µm ± 5.9 µm, while the valley width decreased to 267.5 µm ± 17.6 µm. With a reduction of 19.1% for the height difference and 18.8% for the valley width, these results closely correspond to the globally estimated shrinkage of the cylindrical diameter. Therefore, it is reasonable to assume that the decrease in both overall size and microstructure features is consistent.

#### 7.2.2. Silicone Filling

Compared to Gelafix filling, Dragon Skin 10 Fast filling exhibited a significantly lower shrinkage, measuring less than 0.1% according to the manufacturer’s data sheet (https://www.kaupo.de/shop/out/media/DRAGON_SKIN_SERIE.pdf, accessed on 19 March 2024). Our own assessment, involving the creation of a mold from master targets followed by casting with Dragon Skin 20A and subsequent measurement of the average microstructure dimensions, yielded an estimate of less than 1% change in size attributable to the process. Moreover, the silicone retained its softness even after a five-month period and exhibited no signs of deformation. Figure 16b illustrates the silicone filling for the initial resin-printed mold, while Figure 16e depicts the silicone of the second version resin-printed mold. Notably, the silicone phantom produced from the Alpine3D GmbH SLA service-printed resin mold is devoid of artifacts.

Only for the whorl fingerprint type did we notice a problem with the ridge line depth. The recorded fingerprint impressions showed two vertical.

The ridge line depth and width for the left loop fingerprint class of the Alpine3D GmbH phantom were measured to be 37.5 µm ± 4.5 µm and 265.0 µm ± 18.0 µm, as can be seen in Figure 20. Additionally, we checked the second fingerprint, which is of the arch fingerprint class and is located on the back side of the target cylinder. The ride line depth was 41.2 µm ± 5.0 µm, and the valley width was 294.5 µm ± 31.4 µm.

Furthermore, we also analyzed the phantom created from the Elegoo Saturn 2-8K-printed mold and the Dragon Skin 20 filling. The maximum ridge-valley depth was about 128.2 µm ± 9.4 µm, and the estimated valley width was 370.1 µm ± 36.9 µm.

### 7.3. CNC Machining of Aluminum Master Target and Filling

Manufacturing the checkerboard master target presented significant hurdles that ultimately prevented achieving the intended design and target tolerances.

In an attempt to overcome these challenges, the length of the checkerboard structure was shortened by 20 mm along the cylinder axis. Additionally, the number of milling cycles was reduced, leading to a final depth shallower than the targeted 70 µm. Unfortunately, even these measures were insufficient to prevent tool breakage and complete the milling process without tool replacement.

The master target variant with an artificially generated fingerprint, and the three Ronchi variants were successfully produced using the manufacturing technology.

We did not encounter problems in the mold creation step nor the casting step. Figure 21 shows the surface of the silicone phantoms under microscopy for both the Ronchi rings (Figure 21a) and the fingerprint (Figure 21b) master targets. The ridge lines have a clear separation from the valleys, and no artifacts could be observed.

Furthermore, the evaluation of the concentric Ronchi pattern (see Figure 7b and Figure 22a,b) with the profilometer showed a good preservation of the master targets structure with a structure depth of 85.1 µm ± 4.3 µm (see Figure 22a) and 529.6 µm ± 13.3 µm (see Figure 22b).

The Fraunhofer Institute conducted roughness measurements on both the blank master targets, recording Rq as 1.2 µm, and the sandblasted master target, measuring Rq as 2.5 µm. These findings demonstrate that both variants fall within the typical Rq range of 1–10 µm for pattern projection-based 3D sensors.

Moreover, the height difference between the ridge and valley of the fingerprint sample is determined to be 75.2 µm ± 2.0 µm, while the valley width measures 320.8 µm ± 10.6 µm, as can be seen in Figure 23.

### 7.4. Overview

Table 3 furnishes an exhaustive summary of the various manufacturing processes employed in this study, encompassing their associated costs, manual labor requirements, and the presence of artifacts in the final products.

From a financial standpoint, both CNC machining of the aluminum master target and laser engraving of the aluminum half-pipe emerge as the most expensive methods, each entailing an expenditure of approximately 1000€ for producing a new target type. Conversely, in-house 3D resin printing processes prove to be the most economical, primarily due to the relatively low cost of printing resin and filling materials.

Regarding manual labor, the CNC machining of the aluminum half-pipe target, coupled with the subsequent molding of the silicone negative and silicone phantom process, demands the most manual involvement owing to its multi-step nature. Manufacturing a new phantom with a different fingerprint necessitates two consecutive curing periods, alongside intricate CNC machining of detailed ridge line patterns, involving small cutters, thus resulting in the longest production time. Conversely, a faster production approach with reduced manual labor entails the printing of 3D resin molds, subsequently filled with a suitable filling material, in our case either Gelafix or silicone. This process only necessitates one curing period, although the printing process itself must be conducted slowly to prevent the introduction of artifacts. However, by amalgamating two fingerprint patterns on one phantom, production speed effectively doubles while the cost is halved. Direct laser engraving of elastomer or silicone involves the least manual labor. In the case of in-house-produced silicone plates, labor is involved in creating the silicone plates, although these can be swiftly manufactured by producing larger plates cut into appropriately sized pieces.

Regarding artifacts, CNC machining of the aluminum half-pipe target yields an error-free master target. Subsequent phases for creating a silicone negative and a silicone phantom do not introduce any artifacts. Conversely, the laser engraving of the aluminum half-pipe process yields noticeable artifacts in the form of offsets in the ridge line pattern, evident in both the aluminum mold and the silicone fillings, which also manifest in the rolled recordings of the phantom. The ES2 Elegoo Saturn 2-8K resin printer used for the first version of the 3D-printed resin fingerprint mold generates numerous artifacts such as small holes and a decrease in ridge depth towards the edge of the mold. However, refinement of the mold design, combined with SLA printing services, produces artifact-free molds. Additionally, during the filling process, no artifacts are introduced. Direct laser engraving of elastomer plates yields varying ridge line widths visible under the microscope but with no effect on the resulting fingerprint pattern. Interestingly, minor ridge line offsets are visible during the laser engraving of in-house-produced silicone plates under the microscope and on the rolled recordings. No artifacts were found on the purchased artificial skin silicone plates.

The handling of finger-like silicone phantoms for rolled recordings proved highly effective. Dragon Skin 10 Fast elasticity closely mimics the behavior of human skin, surpassing Gelafix and Laserline EPDM CSX 5K elastomer. While fresh Gelafix initially behaves similarly to silicone, its handling performance deteriorates rapidly due to hardening and shrinking.

## 8. Results—Fidelity of 3D Targets

### 8.1. NFIQ 2 Analysis of Phantoms and Synthetic Fingerprints

To assess the visual quality of the generated 2D synthetic fingerprint images, we employ the NFIQ 2 tool version 2.2.0. We generated NFIQ 2 scores for both the synthetic images and rolled impressions collected from the corresponding finger phantoms, enabling a direct comparison of visual quality.

Table 4 summarizes the NFIQ 2 measurements for rolled impressions of the fingerprint phantoms (recorded with Greenbit Dactyscan 84c) and their corresponding 2D synthetic representations. Each value represents the average score obtained from multiple recordings of phantoms produced using the same methodology (refer to Table 3 for methodology abbreviations).

It is noteworthy that certain methodologies exhibited no significant difference between the synthetic template and the fingerprint phantom. In some cases, such as with Print ES2, there was a decrease in the quality score, but for all others, the scores remained comparable or even result in an improvement for the phantoms. However, it is important to highlight that all obtained scores fall within the range of medium quality. It is worth noting that NFIQ 2 was not specifically designed for this application scenario.

### 8.2. End-to-End Fidelity

#### 8.2.1. Pre-Processing

To ensure the accuracy of end-to-end fidelity assessments, a pre-processing was conducted to identify optimal scaling values for correcting potential shrinkage and distortions in the 2D to 3D warping process. The results are presented in Table 5.

Note the column $\sigma_{score}$, which describes the standard deviation of the template comparison score values for the ten best scaling options, averaged over all individual recordings for a given phantom type. If the standard deviation is zero, all ten scaled images for all recordings of the fingerprint phantom type have the same template comparison score, which is in this case very likely the high score of 1000 points.

These optimal scaling values were applied to correct recorded rolled fingerprint samples before conducting template comparison assessments.

#### 8.2.2. Matcher Comparison

Following the application of optimal scaling factors, we assessed the end-to-end fidelity of the synthetic finger phantoms using three distinct matcher methodologies: the commercially available automated fingerprint identification system Idkit, the NBIS distribution employing Mindtct and Bozorth3, and a combined approach utilizing FingerNet for minutiae extraction and SourceAFIS for minutiae matching. In this context, it should be noted that the comparison scores obtained from the three systems are not directly comparable with each other since IDKit operates on a closed interval [0,100], whereas both of the other algorithms use an open interval [0,∞).

The results of the matcher comparison are presented in Table 6. Each column represents the average template comparison scores obtained by a specific matcher across all fingerprint phantoms produced using the corresponding methodology (refer to Table 3 for methodology abbreviations). The rows represent different phantom production methodologies and finger types within each methodology.

For the fingerprint phantoms, the laser-engraved silicone plate phantoms consistently exhibited the highest scores across all matchers, followed by the laser-engraved aluminum half-pipe mold filled with silicone. Note that phantoms produced with the Alpine GmbH SLA print service have high end-to-end fidelity scores, except for the phantom with the whorl pattern when using the Idkit matcher. We noticed a problem with the ridge line depth for the whorl fingerprint phantom while creating the fingerprint impressions that led to two vertically stretched regions where the fingerprint pattern is obstructed.

### 8.3. Intra-Class Variability of the 3D Target Impressions

Beyond evaluating the end-to-end fidelity, we investigated the intra-class variability of individual fingerprint phantoms, gauging the consistency of recordings captured from the same phantom type using the three matcher methodologies introduced in Section 5.2.

The results of this analysis are presented in Table 7. Similar to the end-to-end fidelity table, each column displays the average intra-class variability scores obtained by a specific matcher across all fingerprint phantoms produced using the corresponding methodology. As before, rows represent different phantom production methodologies and specific finger types within each methodology.

Methodologies that score high on the end-to-end fidelity test depicted in Table 6 also score high on the same fingerprint phantom type intra-class test depicted in Table 7.

Only the elastomer material as well as the Gelafix-filled 3D-printed resin mold had scores significantly lower than the other manufacturing methods considering the Idkit matcher. The template comparison scores using the Bozorth3 matcher do not provide a clear separation between processing methodologies and furthermore, the Bozorth3 matcher strongly disagrees for the laser-engraved aluminum half-pipe filled with silicone phantom with the other matchers. The phantom had the second lowest score for the Bozorth3 matcher, a perfect score for the idkit matcher, and the highest score of all phantoms for the SourceAFIS matcher. Regarding the SourceAFIS matcher, the Gelafix-filled 3D-printed resin mold scored worse than all the other methodologies. The other methodologies fall into a similar range, with the exception of the laser-engraved aluminum half-pipe filled with silicone phantom, which outperformed all other methodologies.

To broaden the scope, we extended the assessment to encompass diverse manufacturing methodologies for a given fingerprint pattern. Table 8 presents the intra-class variability for phantoms with the same fingerprint type but produced using different methodologies. We compare phantoms produced using the aluminum half-pipe filled with silicone with direct laser engravings on both silicone substrates, showing comparable intra-class variability across matchers. Notably, phantoms created using a 3D-printed resin mold with both silicone and Gelafix filling materials displayed generally lower intra-class variability compared to the laser engraving methodologies. This is consistent with the slightly lower end-to-end fidelity scores of the 3D-printed phantoms when compared to the laser-engraved ones.

The intra-class variability scores over different phantom types show a different behavior for the different matchers. The Bozorth3 matches do not differentiate between the manufacturing methodology since all the comparisons are in a range of 93 to 135 points. In contrast to this, the SourceAFIS matcher clearly finds two clusters. One cluster with a lower score range of 244 to 327 for the comparison of the first version of the 3D-printed resin mold with silicone and Gelafix filling with the CNC-machined master targets and a second cluster ranging from 630 to 880 for the laser-engraved targets. The Idkit matcher also clusters the intra-class variability scores for different phantom types into two clusters, but the score gap between those two is less than for the SourceAFIS case (679 to 770 vs. 956 to 1000).

## 9. Discussion

### 9.1. Applicability

Synthetic physical fingerprints hold significant potential across various applications. Examples of these applications include the following:Test Targets with Ground Truth: Synthetic fingerprint targets, based on digital 3D models, inherently possess a ground truth. Therefore, they serve as valuable tools for testing the accuracy and efficacy of fingerprint scanners in reconstructing this ground truth and furthermore can act as targets in the standardization of fingerprint sensors [67].Data Protection-Compliant Fingerprint Samples: In many countries, strict regulations govern the use of person-related fingerprints [68,69,70]. Synthetic prints provide a solution to circumvent these data protection regulations, enabling for example the publication of fingerprint images without compromising individual privacy.Quality Control: Synthetic fingerprint targets are instrumental in manufacturing processes to ensure the precision and consistency of fingerprint sensors and biometric devices before deployment in real-world scenarios [35]. They facilitate rigorous quality control measures, thereby enhancing the reliability of these devices.Training Humans: Synthetic fingerprint targets are valuable for training operators of forensic fingerprint scanners, particularly for rolled fingerprint captures [71,72,73]. These targets provide physical support and enable hands-on training, ensuring that personnel are adequately trained to handle real-world scenarios effectively.Artificial Intelligence: Synthetic fingerprints can play a pivotal role in training machine learning models for fingerprint recognition algorithms [74,75,76,77,78]. By providing diverse and controlled datasets, synthetic fingerprints contribute to the development of more robust and accurate AI systems for biometric security applications.Presentation Attack: Synthetic fingerprints can also be utilized in simulating presentation attacks, where artificial replicas are employed to assess the vulnerability of fingerprint recognition systems to spoofing attempts [40,79,80]. This allows for the evaluation and enhancement of system security against potential threats.

Overall, standardized fingerprint phantoms are essential for agencies like the Federal Office for Information Security (BSI) and other regulatory bodies. They provide a common benchmark for evaluating the performance of fingerprint recognition systems across different manufacturers and settings. Standardized phantoms ensure consistency and comparability in testing procedures, facilitating the assessment of system accuracy and reliability.

In phantom production, several properties are crucial for ensuring high-quality and reliable recordings. These include low shrinkage over time to maintain consistency in phantom dimensions, appropriate elasticity to facilitate reliable and accurate rolling on the sensor, and minimal surface deformations to prevent distortions in fingerprint images. Phantoms with poor properties may lead to inaccurate or unreliable recordings, compromising the effectiveness of testing and evaluation procedures.

### 9.2. Generating 3D Targets

The comparison of manufacturing methodologies for the creation of fingerprint phantoms elucidates distinct advantages and disadvantages inherent in each approach. Direct laser engraving on Laserline EPDM elastomer presents an economically feasible and expedient method. However, resulting phantoms may exhibit inconsistencies in ridge line thickness and stiffness, potentially impeding their compatibility with certain sensors. Conversely, silicone plates engraved with lasers offer uniform ridge thickness and clarity, yet fail to mimic the elasticity of human tissue or the flattening of the finger when rolled over sensors. The use of an aluminum half-pipe mold for silicone phantoms amalgamates the precision of laser engraving with the malleability of silicone, albeit at a higher cost and potential introduction of artifacts. Additionally, 3D printing of resin molds provides unparalleled flexibility in design, accommodating complex features such as minutiae, and the potential for non-cylindrical finger shapes, albeit with the caveat of requiring attention when designing the 3D model of the mold to prevent small artifacts.

Laser-engraved phantom structures in elastomers, along with our in-house manufactured Dragon Skin 10 Fast silicone and commercially available Gospire silicone, exhibit the clearest distinction between ridges and valleys. This determination is based on the evident emergence of well-defined ridges in the measured profiles. Nonetheless, these findings may be influenced by the flat geometry of the materials, contrasting with the cylindrical shape of the other fingerprint phantoms.

The Dragon Skin 10-based phantoms exhibit a depth range of 37.5 µm ± 4.5 µm, whereas those produced using the Alpine 3D 3D-printed mold reach up to 146.0 µm ± 7.6 µm for a fresh (Day 0) Gelafix-based fingerprint phantom, fabricated with an in-house 3D resin printer. This height disparity between ridges and valleys is adequate for generating images on contact-based fingerprint scanners. However, it is necessary to assess whether the minimal depth suffices to produce distinguishable structures under varying parameters, such as more applied pressure or the use of even softer materials.

Fingerprint phantoms produced with 3D-printed molds commonly feature narrow and pointed ridges, posing challenges in accurately estimating ridge and valley widths. For rolled fingerprints, the narrow and pointed ridges are not critical, since flattening under pressure increases the contact surface area. Nonetheless, it is worth noting that this ridge line pattern increases the risk of interrupted ridges, potentially caused by small air bubbles within the material.

Despite the intermediary step involving the creation of negative molds from aluminum master targets, an accurate preservation of the initial structure depth (75 µm) to the master targets has been achieved (75.2 µm ± 2.0 µm). However, a drawback of this method is the absence of fingerprint structures on the phantom’s fingertip part. The characteristics of the ridges lie somewhere between the phantoms created via laser engraving and casting using 3D-printed resin molds.

To put the measured results into perspective, a ridge line width in the range of 200–850 µm can be expected for a human fingerprint [81]. Furthermore, to indicate the magnitude of possible ridge line depths, we measured and evaluated the ridge line depth of a few human fingers, leading to a ridge line depth of ≈100 µm. Overall, in comparison with 3D captures of real fingers, the flatter phantoms created via laser engraving more closely resemble the ridge line shape of real fingerprints.

The effect of skin color is an important consideration in the development of fingerprint phantoms. In our case, the elastomer material is consistently black, while Gelafix can be ordered in different colors, and silicone can be arbitrarily colored using various color pigments.

Notably, pigmentation is not necessary for contact-based recording processes that make use of capacitive, ultrasonic, or thermal sensors because they do not perceive the color. However, in contactless and optical scanner based applications, matching the skin color of the phantom to a wide range of possible skin colors is essential for accurate representation and reliable performance.

Furthermore, one of the challenges in creating high-quality fingerprint phantoms that strongly influence the fidelity of the fingerprint phantoms is the mapping of the fingerprint pattern onto the 3D target shape. There needs to be a balance between realism of the finger shape and repeatability when creating the fingerprint recordings. A complex shape is susceptible to slight changes in the presentation angle and presentation pressure, which change the parts of the fingerprint pattern that are recorded by the contact-based fingerprint sensors and therefore reduces the average fidelity of the target. On the other hand, a too simple shape fails to test the real-world recording settings of the fingerprint devices. We struck for a middle ground by using a cylindrical base shape, either creating directly a phantom with this shape, or for the plate-like phantoms, by placing them onto a wooden cylinder. The cylindrical geometry has the advantage that the projection from the 2D to the 3D shape can be done without any projection errors. Additionally, we added a spherical half sphere fingertip for the 3D-printed molds to allow for a more life-like shape that could be used for contactless fingerprint scanners.

The scalability of phantom production varies across methodologies. Direct laser engraving on elastomer or silicone plates allows relatively straightforward upscaling, requiring only the creation of new target plates and fast laser processing. Conversely, 3D printing of resin molds offers extensive flexibility in its scalability due to its adaptability to various finger shapes, though hindered by the molding process and curing time. Moreover, CNC machining of aluminum master targets presents a more intricate and time-consuming process, less conducive to large-scale production due to specialized equipment requirements and extended production timelines.

Transitioning to the discussion on the usability of fingerprint phantoms across diverse sensor types, it becomes evident that the efficacy of synthetic phantoms hinges significantly on the manufacturing methodology and material properties. Contactless fingerprint sensors, operating through optical imaging techniques, necessitate phantoms with a realistic finger-like appearance to facilitate fair comparisons across sensors utilizing different pre-processing technologies, like color-based [82] or learned fingertip segmentation [78]. While the laser-engraved aluminum half-pipe mold and 3D-printed resin molds enable the creation of phantoms with lifelike geometry, the latter holds a slight advantage in representing human-like finger phantoms due to its ability to project fingerprint patterns onto rounded fingertip areas.

In contrast, slap recordings, commonly employed for swift and convenient fingerprint capture, pose unique challenges for phantom usability. While the laser-engraved silicone plates demonstrate potential for high-quality slap recordings when used in their flat form, issues arise concerning the surface texture and stiffness of Laserline EPDM elastomers. A lot of pressure was required to register an image at the fingerprint sensor. In order to exclude unique difficulties of the GreenBit Dactyscan 84c scanner with the elastomer targets, we tested the Jenetric Livetouch Quattro (2nd gen) Up, Jenetric Livetouch Quattro Integrated Biometrics Five-O as well. For all tested sensors, the same issues arose, rendering the elastomer phantoms ill-suited for such applications.

For rolled recordings, crucial for high-quality fingerprint identification purposes, silicone phantoms stand out as viable options, especially those with a finger-like shape, due to their elasticity, facilitating smooth rolling over sensor surfaces. Although laser-engraved plate phantoms can be adapted for rolled recordings by mounting them on a cylindrical, wooden target holder, challenges persist due to the thin silicone layer and reduced surface area exposure. This is exacerbated even more for the Laserline EPDM elastomer plates because of the same issues as for slap recordings.

When assessing the durability of fingerprint phantoms, variations primarily arise from the choice of materials. The most pronounced impact of material degradation was observed with Gelafix fillings. As illustrated in Table 2, they experienced significant shrinkage of over 17%, as well as hardening over time, rendering them unsuitable for high-quality sensor calibration and standardization purposes. Conversely, silicone exhibited minimal shrinkage, below 1%, and maintained its softness even after six months, establishing it as the optimal material for producing reliable and consistent phantoms.

### 9.3. Fidelity of 3D Targets

The quality of a fingerprint sample can be assessed according to the three aspects of character, fidelity, and utility, as stated in the iso [83]. As a measure of the utility of a fingerprint phantom, we used the template comparison score between the generated 2D fingerprint and the rolled fingerprint recording. Since the generated fingerprint patterns that are used for the phantom creation have a high character (i.e., are of good quality for template comparison) as shown by the NFIQ 2 values of Table 4 and also share those fingerprint patterns over multiple manufacturing methods, we can interpret the differences in measured template comparison scores not only as differences in utility, but also in fidelity. This allows us to interpret the measured template comparison scores as quality indicator of the measured fingerprint phantom.

Furthermore, for the second iteration of the 3D printed resin mold manufactured by the Alpine3D SLA service, we investigated the influence of different finger types on the character and utility (and therefore also fidelity). We used the fingerprint classification scheme introduced by Karu and Jain in [64], which classifies the fingerprints into arch, tented arch, left loop, right loop, and whorl. All generated 2D fingerprints have a NFIQ 2 score between 46 and 59 and can therefore be interpreted as a fingerprint with a good visual quality [84]. Also, the fingerprints recorded from rolling the corresponding phantoms on the GreenBit Dactyscan 84c can be considered good quality because they are in a similar range of 40 to 55. In both cases, the whorl fingerprint pattern was the lowest scoring fingerprint class. This should not be interpreted as a worse performance of the whorl fingerprint class, since the values are calculated only for a single fingerprint pattern. But, since the character for the whorl class is lower than for the other fingerprint classes, we can also expect the utility of the measured samples to be lower. Moreover, we observed a problem with the ridge depth during fingerprint collection. As a result, the end-to-end fidelity scores for the whorl fingerprint class are lower than for the other patterns considering the Idkit (553) and the SourceAFIS (132) matchers. The other fingerprint classes have similar template comparison scores to one another and fall in the range of between 803 and 977 for the Idkit matcher and 245 to 304 for the SourceAFIS matcher. More research is required to investigate whether the lower utility of the whorl fingerprint class is a singular effect due to the combination of a lower character combined with manufacturing problems, or if it is indicative or a finger class depended difference.

Interestingly, we observed a worse performance of the Bozorth3 matcher with the Mindtct Minutiae extractor for the arch and tented arch fingerprint class, even lower than the whorl fingerprint class. For both Idkit as well as FingerNet combined with SourceAFIS, no such effect is visible. Since both other matchers do not agree on the score difference, we assume that this is a matcher inherent effect.

The overall level of the 3D-printed resin mold filled with silicone improved from the first in-house experimentation to the second version (31.4% Bozorth3, 34.8% Idkit, 21.5% SourceAFIS). The major difference between the 3D model of the in-house-produced sample and the second sample is that the first one has a lower ridge depth on the boundary regions when compared to the center ones. The correction of this, as well as the slower and more precise printing process, strongly influences the end-to-end fidelity. Furthermore, the comparison to Gelafix as an alternative filling material revealed that silicone performed better (29.6% Bozorth3, 7.9% Idkit, −0.5% SourceAFIS). We observed a strong shrinking and hardening of the Gelafix material, which also changed its rolling behavior and made it infeasible for durable fingerprint phantom production, which could be the explanation for the worse performance. Nonetheless, Gelafix is a valuable material for fingerprint phantom production in the context of PAD, because it consists mainly of a mixture of water, gelatine and glycerin, which appears human flesh like to contact based fingerprint sensors. The downside of this material is that the phantoms can only be used for a short period of time.

Besides the 3D-printed molds, both the laser engraving as well as the CNC-machining created phantoms with a very high end-to-end fidelity. Only the elastomer samples are an exception to this, which had the lowest score of all manufacturing methods. This is the case because of the stiffness of the material, which introduced problems creating an image using the contactbased fingerprint scanners.

The intra-class variability scores, which measure the template comparison scores between recordings of one phantom type, confirms the usability of the phantoms for qualifying a fingerprint sensor. Only the elastomer material, as well as the Gelafix-filled 3D-printed resin mold, had scores significantly lower than the other manufacturing methods. This implies that those two materials are not well suited for creating repeatable recordings and overlaps with the insight from the end-to-end fidelity observations. The intra-class variability across different phantom types underlines the conclusions made from the intra-class variability assessment for a single phantom type, highlighting the performance of the laser engraving as a manufacturing methodology and confirming the usability of the 3D printed technology.

Regarding the scaling pre-processing, it should be noted that for the case of very good matching fingerprint recordings that achieve a maximal rating, like those created from CNC machining or laser engraving (except elastomer), the predictive power of the scaling values is lower. This is because even a mismatch in scaling can result in a score that is still capped by the high score, therefore creating a plateau for different scaling values. This is reflected by the zero standard deviation.

Nonetheless, the results of the pre-processing provide an interesting insight: Some phantoms have a non-square optimal scaling value. This is especially prominent for the laser engraving aluminum half-pipe mold with silicone filling and the 3D-printed resin molds filled with silicone. We suspect that this is the case because the projection from 2D to 3D was done for a static case, meaning that the rolling behavior is expected to follow the rolling of a fixed diameter cylinder. However, since the silicone is soft enough to emulate the realistic rolling behavior of a finger, which deforms under pressure because of the contact with the sensor surface, the contact surface widens. This leads to a stretched fingerprint recording along the rolling direction, which is what we can observe. Furthermore, the phantom created from the Gelafix filling material had a higher stiffness and therefore has a less pronounced difference between the x- and y-scaling values.

The repeatability of generating fingerprint recordings of the phantoms, although significantly improved when compared to real finger recordings, is still influenced by many factors, including the presentation angle, presentation pressure, and position on the sensor surface. These factors can vary significantly between different user handling the phantoms, and even for the same user, making it difficult to obtain consistent and repeatable results. A robotic arm, on the other hand, could be programmed to follow a precise rolling procedure, ensuring consistent presentation conditions for each recording and eliminating the human factor from the rolling process. This level of precision is essential for obtaining highly repeatable results in fingerprint comparison and more research is required to create this setup.

### 9.4. Overview

Table 9 presents a comprehensive overview of quality features assessed across various manufacturing methodologies for fingerprint phantoms. Each row corresponds to a specific manufacturing method, while columns represent distinct quality metrics evaluated.

Quality metrics encompassed in the analysis include price, labor requirements in the form of how many manual steps are required, phantom quality measured via end-to-end fideltiy (see Section 5.2), interoperability with different sensors (see Section 5.3), consistency in ridge form and depth, presence of artifacts, geometric shape, and longevity.

The table employs directional arrows to indicate the degree or level of each quality metric, ranging from very low/very few (⇊) to very high/very many (⇈).

It is essential to acknowledge that various applications for fingerprint phantoms may prioritize different quality metrics based on their specific requirements. Consequently, the relative importance assigned to each quality metric may vary accordingly. For instance, applications emphasizing cost-effectiveness and scalability may prioritize metrics such as price and labor requirements. On the other hand, applications demanding high-fidelity representation of real fingerprints may place greater emphasis on metrics such as fidelity and ridge consistency. Similarly, the geometric shape is of high significants when thinking about the different recording modalities. Rolled fingerprint recordings require a 3D phantom shape to similate the rolling procedure, while slap fingerprint recordings might only require a flat phantom.

Our findings suggest that different methodologies are suitable for producing phantoms for rolled, slap or contactless fingerprint sensor recording modalities. For rolled recordings, we recommend laser engraving of a 3D mold followed by silicone casting or CNC machining along with silicone casting. For slap recordings, we recommend laser engraving of silicone plates and for contactless recordings, we recommend 3D printing of the mold.

## 10. Conclusions

This work introduces and compares multiple novel manufacturing methodologies for the production of synthetic 3D fingerprint targets, also called phantoms. The goal is to provide reliable, effective, and standardized practices for creating phantoms that can then be used to validate, test, and standardize the fingerprint recognition sensors and systems. Through extensive evaluation, we have assessed the quality, price, and applicability of different target materials such as silicone, Kryolan Gelafix, and elastomer as well as different production techniques like laser engraving, 3D printing, and CNC machining. An overview of price, labor involved, end-to-end fidelity, interoperability, ridge form, ridge consistency, number of artifacts, phantom geometry, and longevity can be found in Table 9.

Through iterative improvement on the synthetic fingerprint, we end up with a set of synthetic fingerprint images for each fingerprint class that is resolution-independent and can be used in the manufacturing process in arbitrary precision. This allows the harnessing of the full potential of the precision of the laser-engraving methodology. The direct laser engraving of flat targets (both silicone and elastomer materials) creates long lasting, accurate representations of the fingerprint pattern. Furthermore, the laser engraving of a aluminum half-pipe allows us to create a highly precise mold that can be filled with various materials to create a 3D-shaped phantom. However, the creation of the mold suffered from variations in the aluminum half-pipe diameter, which led to longitudinal artifacts that propagated through the phantoms to the final recordings. Furthermore, the increased complexity of laser engraving the inside of the hollow half-cylinder significantly increased the costs.

A cheaper alternative to laser engraving is 3D printing. We tested two different mold designs and two different printing approaches—using the in-house ES2 Elegoo Saturn 2-8k printer and our sourcing of the printing to an SLA print service. We find that avoiding furrows in the mold design to not trap liquids and using a transparent resin for reducing undesired interactions between the color particles in the resin, and the silicone filling is important for creating an artifact-free mold. An advantage of the printing approach is that more complicated structures can be produced. For example, this was the only appproach where we could project the fingerprint pattern on the rounded fingertip. Furthermore, we experimented with the two filling materials, silicone and Kryolan Gelafix. Gelafix showed promising behavior shortly after phantom production because it consists mainly of a mixture of water, gelatine, and glycerin, which appears human flesh like to contact-based fingerprint sensors. However, after repeated use, the material deformed and a longitudinal observation revealed significant shrinkage (over 17% after 6 months) and hardening over time. Silicone, on the other hand, stayed elastic and displayed only minimal shrinkage (<1%) and hardening after 6 months. Furthermore, the elasticity of the silicone material (DragonSkin 10 A) is well suited for rolling based or recordings.

The last manufacturing methodology we tested was CNC machining of the target masters, followed by casting a silicone negative and then the silicone phantom. This process was both labor intensive and expensive, but resulted in very precise phantoms.

Following the manufacturing process, we recorded fingerprints using the created phantoms. We compared NFIQ 2 scores of the synthetic input images and the resulting fingerprints and observed no significant difference between them. Also, all scores fall roughly in the same range.

The next step was investigating the similarity of the resulting recorded fingerprints from the synthetic input images. For this, we calculated the end-to-end fidelity scores using two open-source (NBIS, FingerNet + SourceAFIS) frameworks and one commercial one (Idkit). The scores showed that the laser engraving of silicone plates achieved the best scores, while laser engraving of the elastomer plates the worst. This was due to the material stiffness, which hindered the recording process. But also laser engraving of the aluminum half-pipe and CNC machining of the master target resulted in good end-to-end fidelity scores, while 3D printing with a silicone filling led to reasonable good scores. Furthermore, we tested the intra-class variability, showing that the phantoms can be used to reliably create the same fingerprint reocrding, except for the phantom created using the Gelafix material. Additionally, we looked at the inter-operability of the intra-class variability for the methodologies that used the same synthetic input image, highlighting the great interoperability between the laser-engraved phantoms and the good interoperability between the printed and CNC machined phantoms.

As a result, our findings suggest that employing a combination of laser engraving on aluminum half-pipe molds followed by silicone casting or CNC machining of an aluminum master target, along with a two-stage silicone casting process, strikes the optimal balance between cost, production efficiency, and accuracy in generating finger-like shaped fingerprint phantoms for contact-based fingerprint devices. These phantoms serve as valuable tools for validating sensor performance in rolled recordings.

In the context of validating fingerprint sensors for slap recordings, our research indicates that direct laser engraving of flat silicone targets delivers superior performance across all evaluated parameters.

For contactless fingerprint devices, our study recommends utilizing 3D printing of resin molds followed by silicone filling to achieve the most favorable combination of attributes.

## Figures and Tables

**Figure 1 sensors-24-02847-f001:**
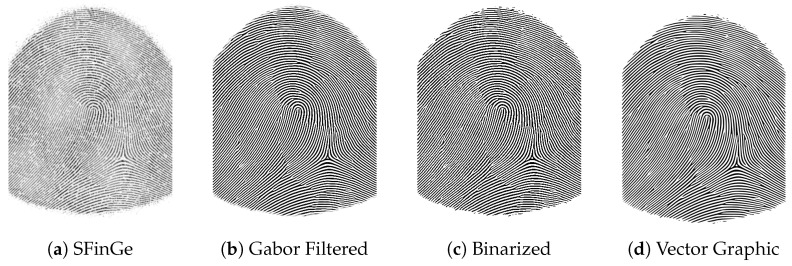
Synthetic fingerprint generation with SFinGe output (**a**), applied Gabor filter (**b**), applied thresholding algorithm (**c**), and finally, path traced and converted to a vector graphic (**d**).

**Figure 2 sensors-24-02847-f002:**
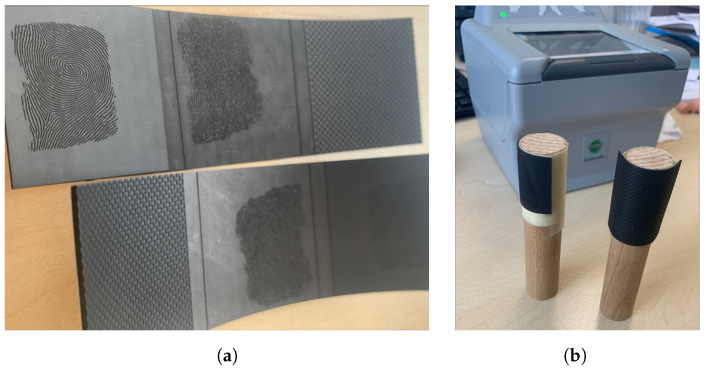
Laser-engraved elastomer targets. (**a**) Laser-engraved elastomer stripes. Stripe with 0.95 mm thickness on top, stripe with 1.42 mm thickness below. (**b**) Exemplary elastomer stripes applied to the wooden target holder.

**Figure 3 sensors-24-02847-f003:**
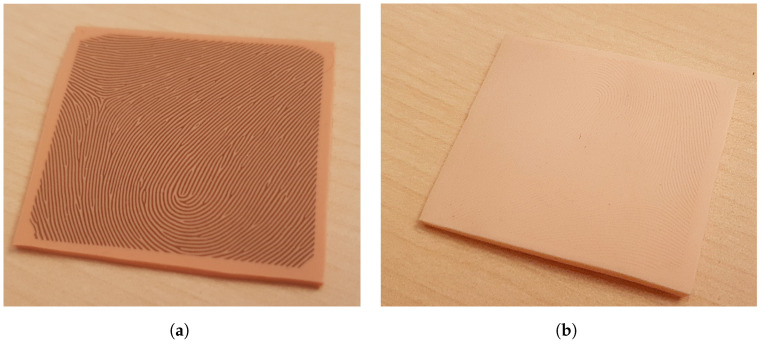
Silicone plates used for laser engraving. (**a**) Silicone plate from the company Gospire used as a training skin for tattoo artists. (**b**) Silicone plate created in-house with Dragon Skin 10 Fast.

**Figure 4 sensors-24-02847-f004:**
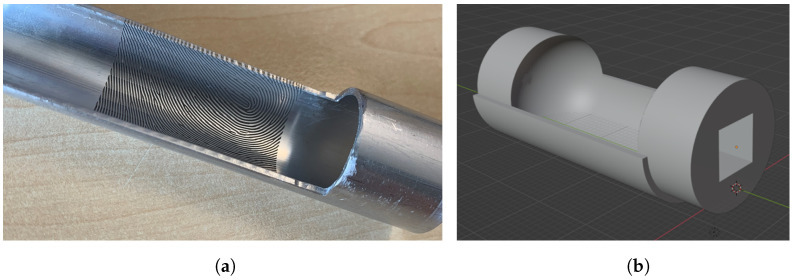
Aluminum half pipe mold. (**a**) Aluminum half-pipe mold with laser engraving. (**b**) Plug for filling the mold with silicone.

**Figure 5 sensors-24-02847-f005:**
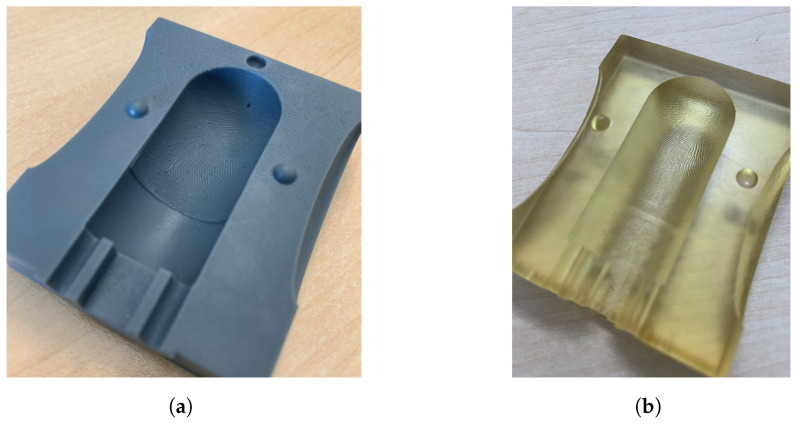
Resin-printed mold halves. (**a**) Printed using the ES2 Elegoo Saturn 2-8K resin printer. (**b**) Printed using the Alpine3D GmbH SLA service.

**Figure 6 sensors-24-02847-f006:**
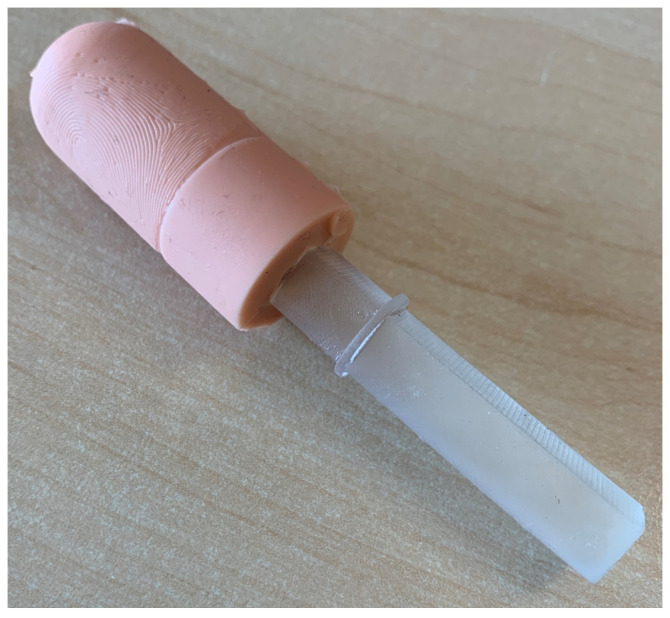
Silicone target made from 3D-printed resin mold.

**Figure 7 sensors-24-02847-f007:**
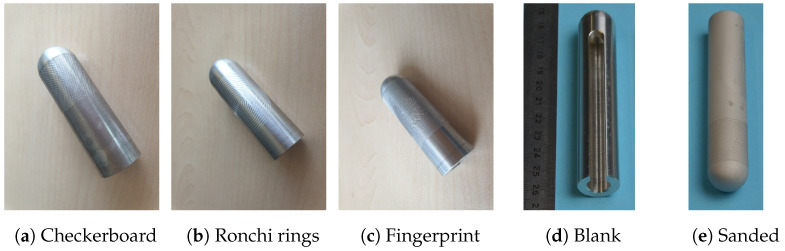
CNC-machined master targets.

**Figure 8 sensors-24-02847-f008:**
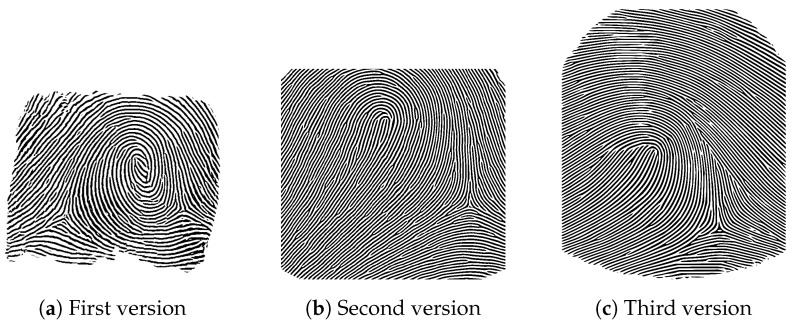
Synthetic fingerprint results used for the following: laser-engraved elastomer samples (**a**), both laser-engraved silicone samples and the laser-engraved aluminum half-pipe mold (**b**), and the first version of the 3D-printed resin mold (**c**).

**Figure 9 sensors-24-02847-f009:**
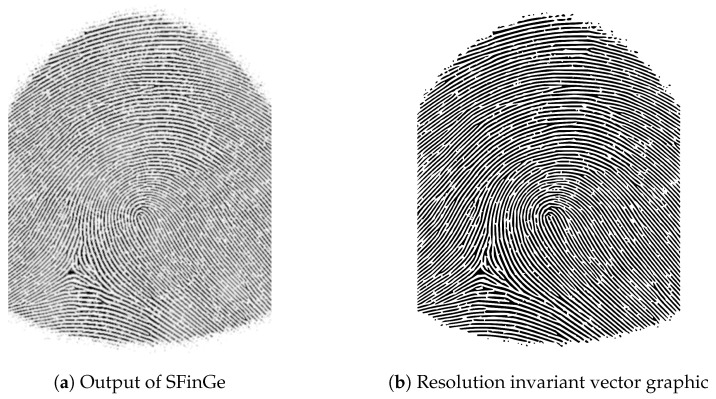
Upsampling of synthetic fingerprint image used for 3D-printed second-generation resin mold.

**Figure 10 sensors-24-02847-f010:**
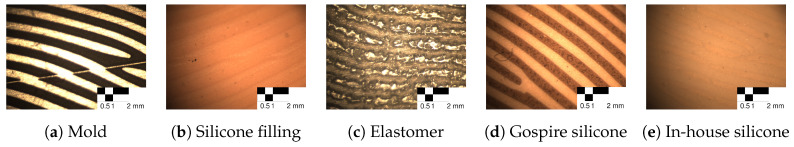
Laser-engraved phantoms (**b**–**e**) and mold (**a**). Phantom (**b**) is the silicone filling of the aluminum mold (**a**), and the other images are direct laser engraving on elastomer (**c**) and silicone (**d**,**e**).

**Figure 11 sensors-24-02847-f011:**
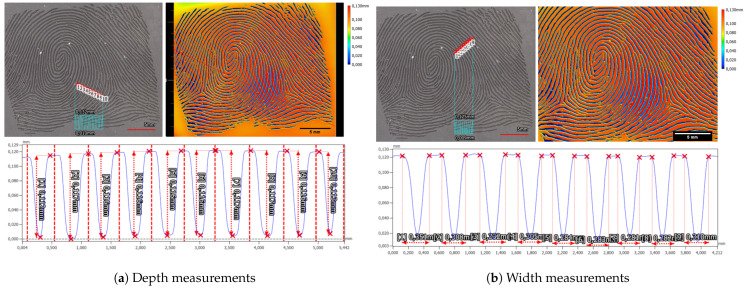
The figure shows the top-down view on the synthetic fingerprint engraved in the elastomer with the corresponding color coded height profile. Scale bar 5 mm.

**Figure 12 sensors-24-02847-f012:**
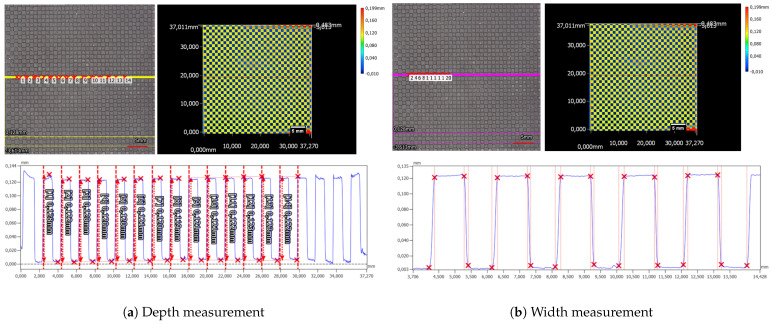
The figure shows the top-down view on the checkerboard structure engraved in the elastomer with the corresponding height profile color coded. The yellow/magenta line highlights the area of the height profile in the bottom plot, while the read dashed line indicates the same area in the height map. Within the plot, the increasing and decreasing shoulder and the upper and lower plateau of the structure were selected by hand. Scale bar 5 mm.

**Figure 13 sensors-24-02847-f013:**
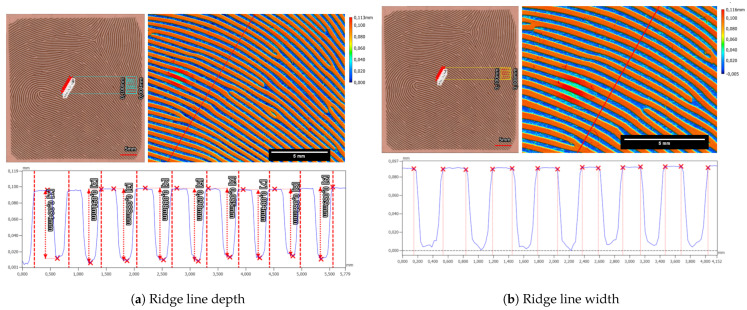
Top-down capture of the Gospire silicone plate. The cyan/yellow line in the upper left image highlights the area where the height profile is measured and the red line the corresponding area in the height map. Scale bar 5 mm.

**Figure 14 sensors-24-02847-f014:**
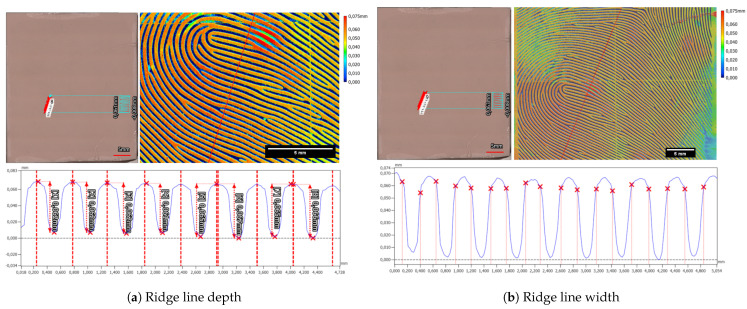
Top-down capture of the in-house created Dragon Skin 10 Fast silicone plate. The cyan/yellow line in the upper left image highlights the area where the height profile is measured and the red line the corresponding area in the height map. Scale bar 5 mm.

**Figure 15 sensors-24-02847-f015:**
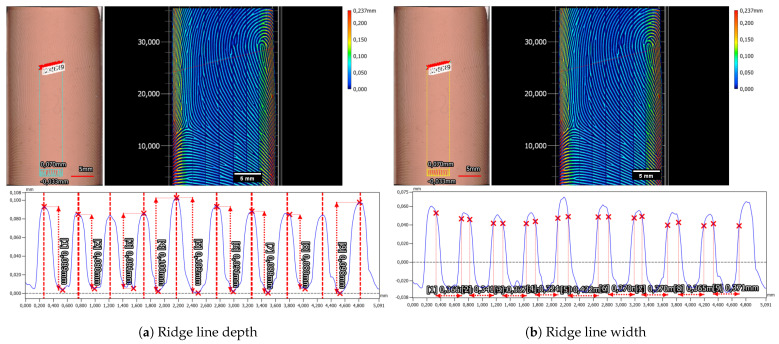
Top-down capture of an in-house Dragon Skin 10 Fast synthetic fingerprint sample created by casting the silicone in an aluminum half-pipe with laser-engraved fingerprint structure. The cyan/yellow line in the upper left image highlights the area where the height profile is measured and the red line the corresponding area in the height map. Scale bar 5 mm.

**Figure 16 sensors-24-02847-f016:**
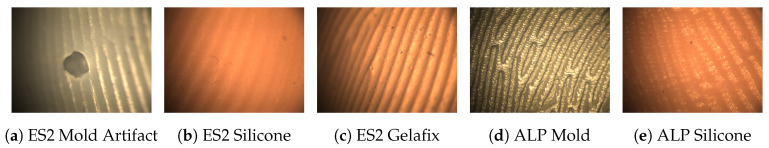
3D-printed resin molds and fingerprint phantoms. Created with the ES2 Elegoo Saturn 2-8K resin printer (ES2) or the Alpine3D GmbH SLA service (ALP).

**Figure 17 sensors-24-02847-f017:**
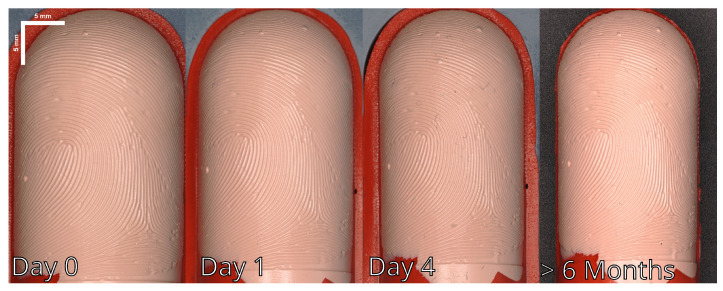
Top-down captures of created Gelafix-based fingerprint targets on different days. From **left** to **right**, day 0, day 1, day 4, and after approximately 6 months. The corresponding diameters of the fitted cylinders are summarized in Table 2. Scale bar 5 mm.

**Figure 18 sensors-24-02847-f018:**
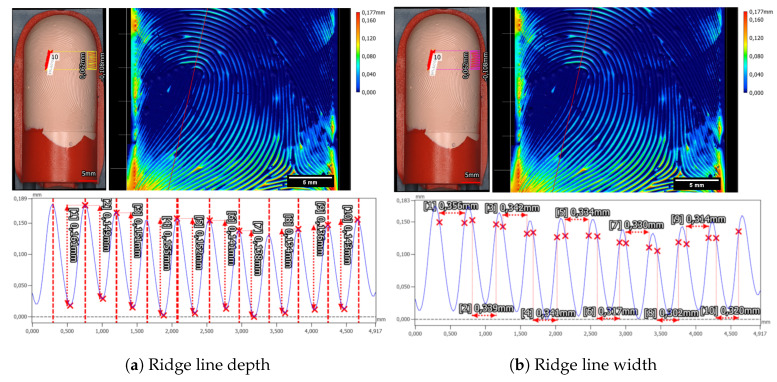
Top-down capture of a fresh Gelafix sample created using the in-house ES2 Elegoo Saturn 2-8K resin printer. The yellow/purple line highlights the area selected for measuring the height profile, which can be seen in the height profile via the dashed red line.

**Figure 19 sensors-24-02847-f019:**
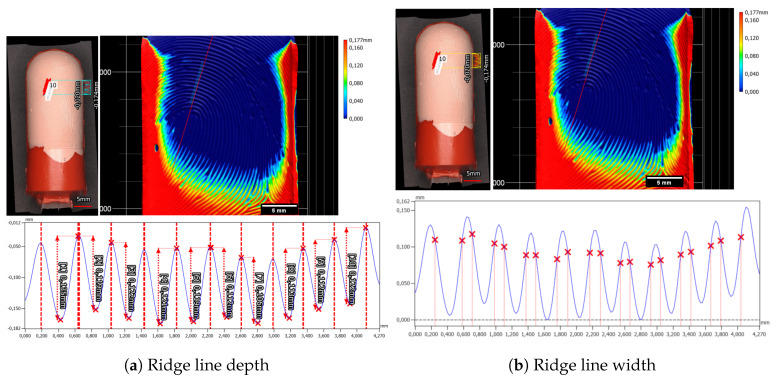
Top-down capture of an approximate 6-month-old Gelafix sample created using the in-house ES2 Elegoo Saturn 2-8K resin printer. The cyan/yellow line, which can be seen in the height profile via the dashed red line. highlights the area selected for measuring the height profile.

**Figure 20 sensors-24-02847-f020:**
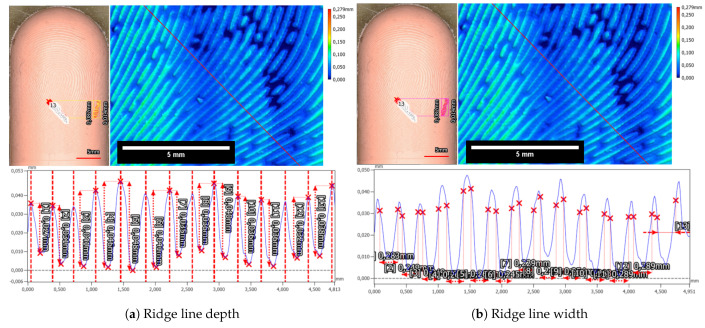
Finger phantom cast with Dragon Skin 10, while using the 3D resin-printed mold by Alpine 3D. The yellow/purple line highlights the area selected for measuring the height profile, which can be seen in the height profile via the dashed red line.

**Figure 21 sensors-24-02847-f021:**
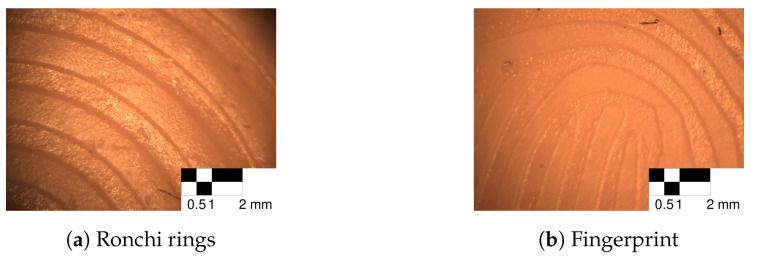
Fingerprint phantoms made from CNC-machined aluminum master targets.

**Figure 22 sensors-24-02847-f022:**
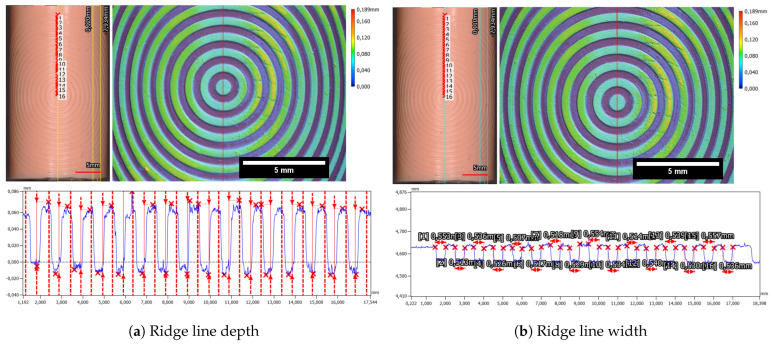
Top-down captures of the Dragon Skin 10 Fast-based fingerprint targets with a concentric Ronchi pattern created from the CNC-machined aluminum master target. Along the yellow/cyan line, the height profile is measured, which can be seen in the height profile via the dashed red line.

**Figure 23 sensors-24-02847-f023:**
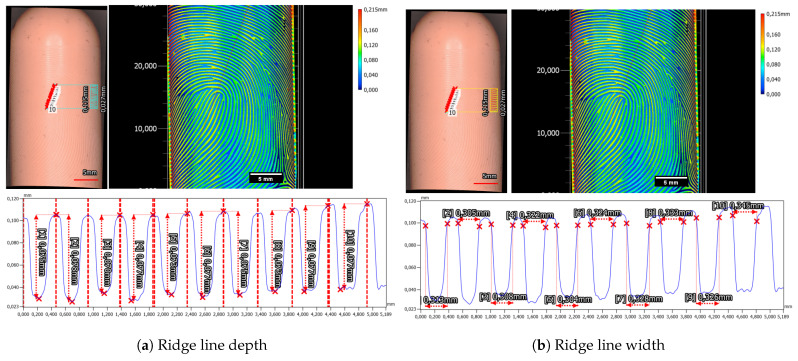
Dragon Skin 10 Fast-based fingerprint targets with a synthetic fingerprint created via a negative mold taken from the aluminum milled master targets. The height profile along the cyan/yellow line is taken to measure the ridge line depth and width, which can be seen in the height profile via the dashed red line.

**Table 1 sensors-24-02847-t001:** Dragon Skin vs. human skin properties.

	Dragon Skin 10 Fast [44]	Dragon Skin 20 [44]	Human Skin
Density [kg/m^3^]	1070	1080	1250 [45]
Pot Life [min.]	8	25	-
Cure Time	75 min	4 h	-
Hardness [Shore A]	10	20	20–41 [46,47]
Shrinkage [m/m]	<0.001	<0.001	-

**Table 2 sensors-24-02847-t002:** This table shows the shrinkage of a fingerprint target made of Gelafix.

Age	Measured Cylindrical Diameter in mm	Shrinkage in %
Day 0	19.29	
Day 1	17.61	8.71
Day 4	16.36	15.19
>6 Months	15.97	17.22

**Table 3 sensors-24-02847-t003:** Overview of manufacturing processes. High Manual Labor implies two casting steps, Medium one casting step, and Low no casting required. Alum stands for aluminum, Print stands for the 3D printing of resin mold methodology, and ES denotes printed with the in-house ES2 Elegoo Saturn 2-8K, while ALP denotes printed using the SLA printer service of Alpine3D GmbH. Laser stands for laser engraving, and HP stands for the half-pipe mold. CNC Alum stands for the CNC Machining of Aluminum Master Target process.

Methodology	Price	Manual Labor	Artifacts	Valley Width [µm]	Ridge Depth [µm]
CNC Alum	setup + ≈1000€	High	Minimal	530±13	86±4
Laser Alum HP Silicone	≈8000€ + 1000€	Medium	Offset of ridge lines	364±20	90±7
Laser Elastomer	≈200€	Low	Varying ridge thickness	304±19	115±1
Laser Silicone Gospire	≈200€	Low	Minimal	340±13	87±3
Laser Silicone In-house	≈200€	Medium	Small offset of ridge lines	303±8	64±2
Print ES2 Gelafix	≈400€ + <10€	Medium	Air bubbles in phantom and small holes in mold	330±18	146±8
Print ES2 Silicone	≈400€ + <5€	Medium	Small holes in mold	370 ± 37	128 ± 9
Print ALP Silicone	≈3000€ + 50€	Medium	Minimal	265±18	38±5

**Table 4 sensors-24-02847-t004:** NFIQ 2 measurements for rolled impressions of fingerprint phantoms recorded with Greenbit Dactyscan 84c (Phantom) and their corresponding 2D synthetic fingerprint (2D Synth). Values are averaged over multiple recordings for a given phantom creation methodology. Abbreviations of Methodology follow the scheme introduced in Table 3.

Methodology	2D Synth	Phantom
CNC Alum	56	53
Laser Alum HP Silicone	44	52
Laser Elastomer	53	61
Laser Silicone Gospire	44	47
Laser Silicone Inouse	44	42
Print ES2 Gelafix	56	36
Print ES2 Silicone	56	37
Print Alpine Silicone	54	50
-Arch	53	54
-Tented Arch	54	55
-Left Loop	59	50
-Right Loop	57	52
-Whorl	46	40

**Table 5 sensors-24-02847-t005:** Optimal scaling values for the recorded rolled fingerprint samples created with the methodology written in the Methodology column in the horizontal direction (x) and the vertical direction (y). Abbreviations of Methodology follow the scheme introduced in Table 3.

Methodology	x	y	$\sigma_{\mathrm{score}}$
CNC Alum	0.57	0.58	0.0
Laser Alum HP Silicone	0.91	1.03	0.0
Laser Elastomer	0.99	1.02	8.3
Laser Silicone Gospire	0.93	1.04	0.0
Laser Silicone Inouse	0.86	1.00	0.0
Print ES2 Gelafix	0.79	0.83	13.9
Print ES2 Silicone	0.68	0.80	13.7
Print Alpine Silicone	0.64	0.88	20.7

**Table 6 sensors-24-02847-t006:** End-to-end fidelity for phantoms measured via template comparison scores of Idkit (Idkit), Mindtct + Bozorth3 (Nbis), and FingerNet + SourceAFIS matcher (FinSource) and their average (Avg). Abbreviations of Methodology follow the scheme introduced in Table 3. The finger types (-Arch, -Tented Arch, …) are sub-categories for the phantoms printed using the SLA printer of Alpine3D GmbH.

Methodology	Avg	Idkit	Nbis	FinSource
CNC Alum	481	1000	158	285
Laser Alum HP Silicone	593	1000	104	674
Laser Elastomer ^1^	93	112	44	123
Laser Silicone Gospire	771	1000	186	1126
Laser Silicone In-house	689	1000	158	908
Print ES2 Gelafix ^2^	275	570	54	201
Print ES2 Silicone ^2^	295	615	70	200
Print ALP Silicone ^2^	388	829	92	243
-Arch	421	904	53	304
-Tented Arch	373	803	71	245
-Left Loop	465	977	130	289
-Right Loop	430	906	137	246
-Whorl	252	553	72	132

^1^ The elastomer phantom was not rolled but placed on the acquisition area, and the wooden finger model was rolled with pressure over the elastomer phantom to simulate the rolling process. ^2^ The phantoms created from the 3D-printed molds are the only phantoms with a fingerprint pattern that is partially on the rounded fingertip area.

**Table 7 sensors-24-02847-t007:** Intra-class variability for phantoms measured via template comparison scores of Idkit (Idkit), Mindtct + Bozorth3 (Nbis), and FingerNet + SourceAFIS matcher (FinSource) and their average (Avg). Abbreviations of Methodology follow the scheme introduced in Table 3. The finger types (-Arch, -Tented Arch, …) are sub-categories for the phantoms printed using the SLA printer of Alpine3D GmbH.

Methodology	Avg	Idkit	Nbis	FinSource
CNC Alum	569	1000	277	429
Laser Alum HP Silicone	624	1000	127	745
Laser Elastomer ^1^	537	778	207	625
Print ES2 Gelafix ^2^	363	727	123	238
Print ES2 Silicone ^2^	514	954	190	408
Print ALP Silicone ^2^	538	985	173	457
-Arch	545	1000	145	491
-Tented Arch	518	991	147	416
-Left Loop	561	1000	210	474
-Right Loop	509	959	177	392
-Whorl	557	974	186	512

^1^ The elastomer phantom was not rolled but placed on the acquisition area, and the wooden finger model was rolled with pressure over the elastomer phantom to simulate the rolling process. ^2^ The phantoms created from the 3D-printed molds are the only phantoms with a fingerprint pattern that is partially on the rounded fingertip area.

**Table 8 sensors-24-02847-t008:** Intra-class variability for phantoms from different manufacturing methodologies with the same fingerprint type measured via template comparison scores of Idkit (Idkit), Mindtct + Bozorth3 (Nbis), and FingerNet + SourceAFIS matcher (FinSource) and their average (Avg). Abbreviations of the Methodology follow the scheme introduced in Table 3.

Methodology	Avg	Idkit	Nbis	FinSource
Laser Alum HP Silicone, Laser Silicone Gospire, Laser Silicone In-house	575	966	99	660
Laser Alum HP Silicone, Laser Silicone Gospire	608	1000	119	705
Laser Alum HP Silicone, Laser Silicone In-house	560	956	93	630
Laser Silicone Gospire, Laser Silicone In-house	661	978	125	880
Print ES2 Silicone, Print ES2 Gelafix, CNC Alum	357	708	103	260
Print ES2 Gelafix, CNC Alum	343	679	106	244
Print ES2 Silicone, CNC Alum	411	770	135	327
Print ES2 Silicone, Print ES2 Gelafix	383	764	118	267

**Table 9 sensors-24-02847-t009:** Overview of the results for the investigated quality features for the different phantom manufacturing methodologies. Laser Elastomer stands for the direct laser engraving of elastomer targets, Laser Silicone for both direct laser engravings of silicone plates created in-house and bought from Gospire, Laser Alum. HP for the laser engraving of an aluminum half-pipe with silicone filling, 3D Print Gelafix for the 3D-printed resin mold filled with Kryolan Gelafix, 3D Print Silicone for the combination of both versions of the 3D-printed resin molds filled with silicone, CNC Alum. Silicone stands for the phantoms casted in a two-stage process from CNC machined aluminum master targets. Interop. stands for interoperability, R. Form for ridge form, R. Consistency for ridge-valley width and depth consistency. The arrows indicate very low/very few (⇊), low/few (↓), medium (˷), high/many (↑), very high/very many (⇈).

	Price	Labor	Fidelity	Interop.	R. Form	R. Consistency	Artifacts	Geometry	Longevity
Laser Elastomer	↓	↓	⇊	↑	Wide	˷	Minor	Flat	↑
Laser Silicone	↓	↓	⇈	↑	Wide	↑	Minimal	Flat	↑
Laser Alum. HP	↑	˷	↑	↑	Wide	˷	Major	Cylindrical	↑
3D Print Gelafix	˷	˷	↓	˷	Slim	˷	Major	Cylindrical + Tip	↓
3D Print Silicone	˷	˷	˷	↑	Slim	˷	Minimal	Cylindrical + Tip	↑
CNC Alum. Silicone	↑	↑	↑	↑	Slim	↑	Minimal	Cylindrical	↑

## Data Availability

The data are contained within the article.

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
