# Peer review of "Toward Synthetic Physical Fingerprint Targets"

_sensors, 2024, doi:10.3390/s24092847_

Round 1

Reviewer 1 Report

Comments and Suggestions for Authors

Biometric fingerprint identification was summarized and several techniques were explained for fabrication such as laser engraving, 3D printing, and CNC machining techniques, utilizing different materials. The paper was detailed written and also gave the comparison of the difference. My comments are following:

1) All the figures should give scale bars

2) some latest refs could be cited  in the introductiondoi.org/10.1016/j.sna.2024.115275; 10.1016/j.compscitech.2021.109212; doi/10.1021/acsami.3c12910; 

3) The conclusions should be rewritten and it is not so convinced.

Reviewer 2 Report

Comments and Suggestions for Authors

The work presented in this paper focuses on the development of synthetic 3D fingerprint targets, or phantoms, for the evaluation and validation of fingerprint sensors. The authors propose methodologies that utilize laser engraving, 3D printing, and CNC machining techniques to create phantoms that closely resemble real human fingerprints. This study makes a significant contribution to the field of biometric fingerprint identification by addressing the challenges associated with calibrating and standardizing fingerprint sensors. The methodologies proposed in this paper can be valuable for researchers and practitioners working in the field of biometrics and can pave the way for future advancements in fingerprint recognition technology.

The article provides interesting fundamental results but some details should be improved.

1. Section 8.1. Applicability

There are a list of application of Synthetic, physical fingerprints, please support this examples relevant links on recent research to make sure that your research can be useful in this fields.

2. Table 10 should be added to discussion section. Please, summarize strengths and weaknesses of each technology and based on it propose the best protocole of fingerprint production.

3. Please add to conclusion potential perspectives of the result of research. Conclusion should include main result of research, but there is no results about Shrinkage after storage of fingerprints (fig. 14)

4. Please, don’t include into the name of the fig. this phrase ,, This or the figure shows…,, Cheack all figs.

5. Do you check synthetic fingerprints with real fingerprints after production and after storages? How significant the differences? In abstract you report that you propose methodologies for fabricating synthetic 3D fingerprint targets, or phantoms, that closely emulate real human fingerprints. How was estimate this differences?

6. In abstract to also report that your research offer a unique perspective (line 14). What makes the approach unique? Could you support this thesis with current results?

7. How was estimate intra-class variability? (line 8), please support with results?

8. This research are very complex, therefore it’s rather difficult to find useful for other researcher information. I think if you provide block-scheme of you research it’ll be useful to get information about what kind of materials are used and what kind of them was chosen.

The authors did an excellent job of producing and analyzing fingerprints, leading to a finished manuscript that can be published in a journal, but with corrections and suggested improvements.

Reviewer 3 Report

Comments and Suggestions for Authors

1 The article uses a lot of space to introduce fingerprint generation in related work, but the article only uses a few paragraphs to describe the generation method in the second chapter, which does not reflect the author's innovation in the generation method part.

2 In the last paragraph of the second section of the paper, the article briefly explains the method of synthetic fingerprinting. In this section, it is recommended to add a graphical description of the process to make the method more clear.

3 I think that the synthesis of fingerprint images does not seem to have a certain correlation with the purpose of the article, and I would like to ask the author to further explain the necessity of fingerprint image synthesis.

4 In Chapter 3, the paper describes the synthesis of 3D targets. I think this section should first highlight the innovation of the article, and then give a specific introduction to the synthesis method according to the innovative content.

5 In Table 1 of the paper, the materials used do not seem to be described in the text, and it will be easier for the reader to understand the contents of the table if the table is appropriately introduced in the text.

6 In Chapter 5 of the article, I think that an image should be added to explain the results in more detail. Although some of the results are shown in Figure 7, there is only a limited analysis of these results.

7 The numbers in the schematic diagram of the experimental results are blurry, so it is recommended to increase the image resolution.

Comments on the Quality of English Language

Minor editing of English language required

Round 2

Reviewer 3 Report

Comments and Suggestions for Authors

The authors have addressed my concerns.

Comments on the Quality of English Language

Minor editing of English language required